

# Modeling anaerobic soil organic carbon decomposition in Arctic polygon tundra: insights into soil geochemical influences on carbon mineralization

Jianqiu Zheng[1], Peter E. Thornton[2,3], Scott L. Painter[2,3], Baohua Gu[2], Stan D. Wullschleger[2,3], David E. Graham[1,3]

[1]Biosciences Division, Oak Ridge National Laboratory, Oak Ridge TN, 37931 USA
[2]Environmental Sciences Division, Oak Ridge National Laboratory, Oak Ridge TN, 37931 USA
[3]Climate Change Science Institute, Oak Ridge National Laboratory, Oak Ridge TN, 37931 USA

*Correspondence to*: David E. Graham (grahamde@ornl.gov)

This manuscript has been authored by UT-Battelle, LLC, under contract DE-AC05-00OR22725 with the US Department of Energy (DOE). The US government retains and the publisher, by accepting the article for publication, acknowledges that the US government retains a nonexclusive, paid-up, irrevocable, worldwide license to publish or reproduce the published form of this

manuscript, or allow others to do so, for US government purposes. DOE will provide public access to these results of federally sponsored research in accordance with the DOE Public Access Plan (http://energy.gov/downloads/doe-public-access-plan).

**Abstract.** Rapid warming of Arctic ecosystems exposes soil organic matter (SOM) to accelerated microbial decomposition,

potentially leading to increased emissions of carbon dioxide ($CO_2$) and methane ($CH_4$) that have a positive feedback on global warming. The fate of permafrost carbon is determined in large part by soil moisture, and a significant portion of carbon may thaw in wet, anoxic conditions. Current estimates of the magnitude and form of carbon emissions from Earth system models include significant uncertainties since the models do not explicitly represent anaerobic carbon decomposition. Here we coupled modeling principles developed in different disciplines, including a thermodynamically based microbial growth model for

methanogenesis and iron reduction, a pool-based model to represent upstream carbon transformations, and a humic ion-binding model for dynamic pH simulation to build a more versatile carbon decomposition model framework that can be applied to soils under varying redox conditions. This new model framework was parameterized and validated using synthesized anaerobic incubation data from permafrost affected soils along a gradient of fine-scale thermal and hydrological variabilities across Arctic polygonal tundra. The model accurately simulated anaerobic $CO_2$ production and its temperature sensitivity using data on labile

carbon pools and fermentations rates as model constraints. $CH_4$ production is strongly influenced by water content, pH, methanogen biomass, and presence of competing electron acceptors, resulting in high variability in its temperature sensitivity.
This work provides new insights into the interactions of SOM pools, temperature increase, soil geochemical feedbacks, and resulting $CO_2$ and $CH_4$ production. The proposed anaerobic carbon decomposition framework presented here builds a mechanistic link between soil geochemistry and carbon mineralization, making it applicable over a wide range of soils under

different environmental settings.





# 1 Introduction

The northern permafrost region contains 1400-1800 Pg soil carbon (C), which is more than twice as much C as is currently contained in the atmosphere (Tarnocai et al., 2009; McGuire et al., 2012). Persistent cold and saturated soil conditions have limited C decomposition in this reservoir. However, rapid warming and permafrost thaw exposes previously frozen organic

carbon to accelerated microbial decomposition, potentially leading to emissions of carbon dioxide ($CO_2$) and methane ($CH_4$) that have a positive feedback on global warming (Zimov et al., 2006; Schuur et al., 2015; Schuur et al., 2009). How quickly frozen soil organic matter (SOM) will be mineralized, and how much permafrost C will be released to the atmosphere following thaw is highly uncertain. Earth system models project 27 -508 Pg carbon release from the permafrost zone by 2100 under current climate forcing (Zhuang et al., 2006; Koven et al., 2015; MacDougall et al., 2012; Schaefer et al., 2014), varying by a factor of thirty.

Understanding environmental dependencies of soil organic matter (SOM) decomposition is therefore essential for reducing model uncertainties and improving predictions of future climate change.

Disagreement in projections of the magnitude and timing of carbon release upon permafrost thaw could be due to differences in model structure, model initialization, or parameters used in simulations. Important geochemical and metabolic constraints might

be poorly represented, oversimplified or missing in current biogeochemical models. To assess the impacts of geochemical constraints on projections of C emissions, we examine two mechanisms that substantially affect SOM turnover in permafrost-affected soils. First, rising temperature alters the kinetics of biogeochemical reactions (Segers, 1998). While temperature acts as an implicit control over permafrost SOM decomposition, the response to temperature rise is an empirical function that can vary for different processes (Treat et al., 2015; Koven et al., 2017). Microbial communities also change with temperature,

compounding effects on process rates (Karhu et al., 2014). Models address this temperature effect using simplified response functions and parameters (Tuomi et al., 2008; Xu et al., 2016), which might be highly biased dependent upon model assumptions and original curve fitting techniques. Second, heterogeneity in permafrost thaw and related hydrological responses modulates decomposition rates and partitioning of $CO_2$ and $CH_4$ production (Schädel et al., 2016). Saturated conditions limit $O_2$ diffusion from the surface, favoring anaerobic respiration, fermentation, and methanogenesis over aerobic respiration. Models use different

levels of detail to simulate effects of water saturation (Meng et al., 2012; Xu et al., 2016), however, soil redox status and pH dynamics associated with oxygen depletion are not widely considered (Riley et al., 2011; Meng et al., 2012).

The fate of permafrost C is determined in large part by soil moisture, particularly water saturation caused by ice melting, precipitation, infiltration and runoff (Riley et al., 2011; Elberling et al., 2013; Schädel et al., 2016). Permafrost thaw frequently

creates large areas of soil inundation due to abrupt surface collapse and subsidence (Painter et al., 2013; Walvoord and Kurylyk, 2016), resulting in higher levels of $CH_4$ production via anaerobic decomposition pathways. Although total carbon release under oxic conditions is much higher than under anoxic conditions (Schädel et al., 2016), emissions of high global warming potential $CH_4$ may offset reduced lower $CO_2$ emissions in the absence of oxygen (Lee et al., 2012).

The extent of SOM decomposition and gas emissions depends upon soil redox potential, beyond simply considering $O_2$ concentrations. Thermodynamics predict that alternative electron acceptors such as $SO_4^-$, Fe(III) and $NO_3^-$ could be favored over methanogenesis and drive C mineralization. The Gibbs free energy available to anaerobic microorganisms that degrade simple organic molecules generally increases (becomes less favorable) with increasing pH (Bethke et al., 2011). Notably, Fe(III) reduction is highly proton consuming and becomes less favorable at higher pH (Figure S1). Previous studies identified Fe(III)

reduction as a major process in anoxic Arctic soils (Lipson et al., 2010; Lipson et al., 2013), which can increase pH and favor co-





occurring methanogenesis (Tang et al., 2016; Wagner et al., 2017). However, the influence of iron reduction on methanogenesis rates in different soils is rarely investigated. The reactivity of iron and its interaction with local soil geochemistry impose additional complexity on the controls of methanogenesis.

Despite the importance of anaerobic SOM decomposition for predicting future climate change, it is not explicitly represented in most biogeochemical models. The SOM decomposition regime is predominantly developed under aerobic respiratory conditions (Manzoni and Porporato, 2009). Current models use landscape position or other proxy of $O_2$ concentration to determine the form of C release. Scalars on aerobic respiration (Riley et al., 2011; Lawrence et al., 2015) or empirical ratios of $CO_2$ and $CH_4$ (Koven et al., 2015) are often used to inform partitioning of $CO_2$ and $CH_4$ production. However, methanogenesis and underlying
anaerobic processes are energetically different from aerobic respiration, generating large unresolved uncertainties.

Some models include details of methanogen populations and their interactions with substrates and other environmental factors, but these models still lack representation of aqueous chemistry reactions that would allow pH, redox potential and thermodynamic calculations (Segers and Leffelaar, 2001a, b; Segers et al., 2001; Xu et al., 2015; Grant, 1998, 1999). Instead, pH
and redox potential are treated as status parameters in these models, and empirical functions are applied for simulations. Without underlying mechanisms, a significant error may occur when rate calculations depend heavily upon the choice of a single optimal value and functional form for various reactions. In addition, the response of SOM decomposition to changes in temperature and moisture remains uncertain (Koven et al., 2017; Schädel et al., 2016). Regardless of modeling approach, the mathematical expressions for temperature or moisture responses vary among current decomposition models, and may not hold for anaerobic
conditions since these relationships are mainly derived from aerobic processes. The lack of explicit anaerobic processes and aqueous chemistry in current biogeochemical models limits their predictive powers.

In this study we developed a new anaerobic carbon decomposition model framework with explicit representation of aqueous phase geochemistry. By coupling three different models, including a thermodynamically based microbial growth model, a pool-
based model, and a humic ion-binding model, we built a process rich carbon decomposition model that allows simultaneous thermodynamic and pH calculations. Results from anoxic incubations of permafrost affected soils along a gradient of soil were synthesized to parameterize and validate this new model framework. The main objectives of this study were to (i) examine the role of soil geochemical variables in controlling anaerobic carbon decomposition and carbon release (as both $CO_2$ and $CH_4$); (ii) develop a common set of parameters in the new anaerobic carbon decomposition framework to capture variabilities in $CO_2$ and
$CH_4$ production; and (iii) evaluate model uncertainties in responses to both soil heterogeneity and model parameterization, emphasizing effects of soil saturation, pH and temperature response.

## 2 Materials and methods

### 2.1 Site description and soil incubations

The Barrow Environmental Observatory (BEO) in Utqiaġvik (Barrow) Alaska, USA consists of thaw lakes, drained thaw lake
basins and interstitial tundra with a polygonal landscape of microtopographic features created by ice wedges. As part of the Next Generation Ecosystem Experiments Arctic project (http://ngee-arctic.ornl.gov/), frozen soil cores were collected from different microtopographic positions of Low-centered, Flat-centered, and High-centered polygons (LCP, FCP, HCP) in the wet tundra.





LCPs are characterized by narrow, saturated troughs, raised rims, and wet, sometimes saturated centers (Figure 1) (French, 2007; Liljedahl et al., 2016). We previously performed short-term incubations of LCP soils under anoxic, environmentally relevant conditions to measure rates and temperature sensitivities of $CO_2$ and $CH_4$ production (Roy Chowdhury et al., 2015). FCPs represent transitional polygons with melting ice wedges, minimal rims, moderately dry centers, and disconnected troughs.

Incubations of FCP soils demonstrated both methanogenesis and methane oxidation potential, with high levels of activity at the transition zone (Zheng et al., 2018). Finally, HCPs have well drained centers and low, saturated troughs. Incubations of HCP soils showed significant fermentation, methanogenesis and anaerobic respiration in the saturated troughs (Yang et al., 2016), contrasted with aerobic respiration and minimal methanogenesis in the centers (Roy Chowdhury et al, in preparation). These controlled incubations provided critical information on anaerobic SOM decomposition processes across a gradient of soil with

fine-scale variability in thermal and hydrological regimes. The results facilitate benchmarking and parameterization for fine-scale anaerobic SOM decomposition models.

Incubation datasets from 8 soil cores, divided in 126 soil microcosms associated with 14 treatments (soil microtopographic features × soil layer) were included in this synthesis to represent the microtopographic heterogeneity of polygonal tundra. Soil

cores were previously sectioned into organic, mineral, cryoturbated transition zone (if identified) and permafrost for microcosm incubations. The period of anoxic incubation in these studies ranged from 45 to 90 days with an average of approximately 60 days at field-relevant temperatures of -2, +4 and +8 °C. Cumulative $CO_2$ and $CH_4$ production data were collected at different time intervals during incubations. More details on the microcosm construction, headspace $CO_2$ and $CH_4$ sampling, and rate calculations can be found in the corresponding publications (Roy Chowdhury et al., 2015; Herndon et al., 2015) and datasets

(Zheng and Graham, 2017; Zheng et al., 2016). Changes in exchangeable Fe(II), water extractable organic carbon (WEOC), low molecular weight organic acids, and pH of soil microcosms during anoxic incubation were summarized previously in publications (Herndon et al., 2015) and datasets (Zheng and Graham, 2017; Herndon et al., 2016).

### 2.2 Data processing and statistics

In order to compare the cumulative carbon loss (as both $CO_2$ and $CH_4$) from different polygonal and microtopographic features and different soil layers, measurements from triplicate microcosms were pooled together and fitted with hyperbolic, sigmoidal, exponential or linear functions that best describe the dynamic. The cumulative $CO_2$ and $CH_4$ production within 60 days of anaerobic incubation was directly calculated from each fitted curve and used for descriptive statistical analyses.

Individual curve fitting for each microcosm was used to best represent the rate changes of $CO_2$ and $CH_4$ production. $CO_2$ production followed hyperbolic curves with immediate $CO_2$ release for all LCP, FCP and HCP trough samples. $CO_2$ production from HCP center samples experienced time lags for approximately 10 days for the mineral layer, and 45 days for the permafrost. $CH_4$ production was also associated with varying time lags before reaching maximum rates, and the lag is most profound in HCP samples, between 6 to 20 days. The rate of gas production estimated from hyperbolic curve fitting predicts a continuously

decreasing rate, while sigmoidal curve fitting with an initial delay predicts a maximum rate after the lag time. Here, we used the derivatives of nonlinear curve fitting to calculate initial rates of gas production. For hyperbolic fittings, the maximum rate is calculated at day 0. For sigmoidal fittings, the maximum rate is calculated by setting the third derivative to zero. The temperature dependence was calculated using conventional $Q_{10}$ by taking the ratio of maximum production rates at 8 and -2°C based on triplicate measurements. Data fitting and statistical analyses were conducted and validated using R 3.4.0 (The R Foundation for



Statistical Computing) and Python 3.6.0 (Python Software Foundation) computing environments. A complete list of packages and libraries used here can be found in the following references (Venables and Ripley, 2002; Hunter, 2007; Oliphant, 2007; Sarkar, 2008; Walt et al., 2011; Wickham, 2009; Wickham et al., 2017).

**2.3 Anaerobic carbon decomposition model**

The anaerobic carbon decomposition framework was developed with explicit representation of fermentation, methanogenesis and iron reduction, which were identified as key mechanisms for anaerobic $CO_2$ and $CH_4$ production (Roy Chowdhury et al., 2015; Yang et al., 2016; Zheng et al., 2018). We used a thermodyanmically-based approach (Istok et al., 2010) to represent methanogenesis and iron reduction, with unique microbial growth kinetics incorporated into energy yielding redox reactions. An aqueous phase was introduced into the model to facilitate thermodynamic calculation and redox-reaction associated acid-base

chemistry calculation. Proton exchange was provided by SOM using a humic ion-binding model (Tipping, 1994; Tipping, 1998), allowing calculation of pH dynamic during anaerobic carbon decomposition. Given the difficulties in partitioning SOC into chemically distinct pools, the CLM-CN decomposition cascade (Converging Trophic Cascade) was adopted to simulate discontinuous carbon pools with different complexity (Thornton & Rosenbloom, 2005, Tang *et al.*, 2016, Xu *et al.*, 2015) and to facilitate parameterization of the upstream carbon flow entering aqueous phase dissolved organic carbon (DOC) pool (Figure 2,

process 1). The above model structure was implemented into the open source geochemical model PHREEQC (Charlton & Parkhurst, 2011). A temperature effect was parameterized using the CLM-CN temperature response function (Appendix B, equation B1).

Under anoxic conditions, the hydrolysis of polysaccharides is considered the rate-limiting step for downstream methanogenesis

(Glissmann and Conrad, 2002). The microbial degradation of cellulose is considerably diminished at low temperature (below 15°C), while other polysaccharides, such as starch, or proteins are degraded much faster at low temperature, resulting in the accumulation of organic acids, primarily acetic, propionic and butyric acids (Kotsyurbenko, 2005; Yang et al., 2016). These polymers are degraded through several hydrolysis and fermentation steps to produce highly biodegradable low molecular weight organic acids that fuel microbial mineralization reactions, leading to production of $CH_4$ and $CO_2$. Instead of including multiple

hydrolysis and fermentation steps, we assumed the turnover of DOC into low molecular weight organic acids is a single lumped fermentation process (Figure 2, process 2). Most anaerobic lignocellulose degraders also ferment sugars following hydrolysis, justifying this assumption (Blumer-Schuette et al., 2014).

Redox reactions that consume or produce protons are sensitive to pH. Reactions such as ferrihydrite reduction substantially

increase alkalinity (Appendix A, Reaction A4). Furthermore, the solubility of $CO_2$ and the composition of dissolved $CO_2$ and bicarbonate vary significantly over typical soil pH values, affecting all C mineralization processes. The humic ion-binding model calculates chemical equilibrium using strong or weak binding sites of organic matter with various proton binding constants (Dudal & Gérard, 2004). A simplified parameterization of proton binding is available in the Windermere Humic Aqueous Model (WHAM, Tipping, 1994; Tipping, 1998), which has been extensively calibrated to represent the acid-base chemistry of

"average" humic and fulvic acids, and it was benchmarked with heterogeneous natural organic matter (Atalay et al., 2009), we adopted the WHAM parameterization to represent pH buffering provided by SOM. Therefore, the pH buffering capacity can be directly adjusted by altering the number of proton binding sites, which is assumed to be linearly correlated with total amount of SOM. The influence of pH on biological reaction rates is modulated by bell shaped pH response functions with a single optimal value between pH 5 to 7 (Tang et al., 2016; Xu et al., 2016). The Dynamic Land Ecosystem Model (DLEM) pH response





function (Appendix B, equation B5) is used in this model since it generates the least variation in parameter perturbation tests (Tang et al., 2016).

### 2.4 Model initialization and parameter uncertainty

In this anaerobic carbon decomposition framework, we focused on developing and validating parameters for anaerobic mechanisms leading to $CO_2$ and $CH_4$ production. The kinetic rate constants and microbial biomass growth and decay rates were adopted from former thermodynamically based studies (Istok et al., 2010). Previous long term incubation studies suggest the size of the fast carbon pool is less than 5% of SOC in permafrost affected soils, with a mean turnover rate ($\tau$) between 77 to 150 days at standardized temperature (4-5 °C) (Knoblauch et al., 2013; Schädel et al., 2014). This turnover rate is most relevant to SOM1

($\tau$=14 days) and SOM2 ($\tau$=70 days) in the decomposition model (Thornton & Rosenbloom, 2005; Tang et al., 2016; Xu et al., 2015), thus we start with 1% and 4% of SOC as the initial values of SOM1 and SOM2. Therefore, a lumped fermentation process was assumed to represent the turnover of DOC into low molecular weight organic acids. The fermentation rate can be directly estimated from $CO_2$ production based upon reaction stoichiometry of fermentative conversion of glucose into acetate and further decomposed via acetotrophic methanogenesis or Fe(III) reduction (Appendix A). The maximal production of $CO_2$ is

about 2/3 of the initial carbon. Thus, the fermentation rate ($R_{fer}$) is estimated using the initial $CO_2$ production rate in the incubation experiment and further optimized by fitting with observed $CO_2$ production.

The model was initialized using measurements based upon 10 to 15 g of wet soil incubated in 60 to 70-mL sealed bottles. Total soil organic carbon (SOC), total water (TOTW), total organic acid carbon (TOAC), pH and initial concentration of Fe(II) were

specified in the model based on measurements. The DOC pool in the model was initialized using the measured WEOC, and expressed as a fraction of SOC ($f_{doc}$) (Table S2). The starting biomass of methanogens and iron reducers was assumed to be in the range of $10^{-4}$ to $10^{-7}$ gC/gSOC and further adjusted to fit the observations. Model fitted fermentation rates are summarized in Table S3.

This model is designed as a generic framework to simulate anaerobic carbon decomposition across a range of soil physiochemical conditions. Two types of sensitivity analysis were conducted to evaluate model performance. First, possible bias and variations associated with model initialization variables (soil geochemical attributes) were assessed using perturbation simulations. Variations of ±25% and ±50% were applied to these variables and the resulting changes in cumulative $CO_2$ and $CH_4$ production were evaluated by comparing with reference simulations. This evaluation helps to identify critical measurements

needed for initializing the model. Second, parameters specifically benchmarked in this study and parameters adopted from empirical relationships were also evaluated with perturbation simulations. This test helps to apportion the model prediction uncertainties into different sources, including model input, parameters, or model structure.

### 3 Results

### 3.1 Synthesized soil geochemical characteristics

Soil samples used in this study represent a wide range of SOC content, from 2% to 39%, with the highest SOC found in surface organic layers. In correlation analysis of initial soil geochemical characteristics, both WEOC and TOAC showed strong





correlation with SOC content among examined soil cores and across soil depth (Table 1). Decomposition generated limited changes in WEOC. WEOC represents 0.26% to 2.6% of total SOC among all test soils, and this ratio remained constant before and after anoxic incubations (Figure 3a). Higher incubation temperatures showed minimal effect on the WEOC/SOC quotient, and a temperature response trend remained insignificant. On the other hand, TOAC showed much more dynamic changes among

different soils and different incubation temperatures. TOAC increased by 5% to 175% in organic soils, and 2% to 60% in the transition zone and permafrost from LCP and HCP centers (Figure 3b). In most mineral soils, TOAC drastically decreased by up to 90%, partially contributing to the depletion of WEOC after the anoxic incubations. These results indicate that WEOC was usually in a steady state, while TOAC varied substantially due to microbial activity.

Good correlations between soil organic carbon (SOC, WEOC and TOAC) and soil moisture were found, indicating the importance of soil moisture in controlling carbon substrate availability (Table 1). High organic carbon content and high soil moisture are associated with organic and permafrost soil, while mineral soils are much drier with lower organic carbon content (Figure S2). Fe(II) concentration is measured as a proxy of soil redox potential, and it is most closely related to soil pH (Table 1).

### 3.2 $CO_2$ and $CH_4$ production, temperature response and soil layer effects

Cumulative production of both $CO_2$ and $CH_4$ showed close correlation with soil carbon content (represented as SOC, WEOC and TOAC) on a dry soil mass basis, as well as soil moisture (Table S4). Given the high inter-correlation among these soil attributes (Table 1), we evaluated the relationship between gas production and soil geochemical characteristics on per gram C basis to avoid inter-correlation caused by SOC content (Lee et al., 2012). The maximum values of cumulative $CO_2$ production were 756 and 534 $\mu mol\ g^{-1}$ C at 8 and -2°C, exceeding the median values at corresponding temperature by 5 and 8 times, respectively. The

maximum cumulative $CH_4$ production was 198 $\mu mol\ g^{-1}$ C from the organic layer of LCP center at 8 °C, approximately 123 times the median value among the rest of the samples. Cumulative $CH_4$ production from the same soil was 9.2 $\mu mol\ g^{-1}$ C when incubated at -2 °C, 9 times the median rate among the remaining samples. Cumulative $CO_2$ production showed strong positive correlation with initial soil moisture at both 8 and -2 °C, while cumulative $CH_4$ production only correlated with initial soil moisture at 8 °C (Table 1). Noticeably, significant correlation between production of $CO_2$ and $CH_4$ was found at 8 °C, but not at -

2 °C (Table 1), suggesting $CO_2$ and $CH_4$ production were controlled by different factors at 8 and -2 °C, respectively. Given that cumulative $CO_2$ and $CH_4$ production both had skewed distributions among samples (Figure S3), the significance in the correlation between gas production and moisture would be substantially weaker with exclusion of measurements from the wet organic layer of LCP center.

Initial production rates of $CO_2$ and $CH_4$ varied significantly across organic, mineral and permafrost soil layers ($p<0.001$ and $p<0.001$, respectively, Figure 4). Temperature showed a substantial positive effect on $CO_2$ and $CH_4$ production rates ($p=0.02$ and $p=0.04$, respectively). A significant temperature × soil layer interaction effect was found on $CO_2$ production rate, but not on $CH_4$ production rate (Table S5), suggesting $CH_4$ production might be more sensitive to constraints from additional environmental conditions.


A $Q_{10}$ value was calculated for each condition to further assess the temperature dependency of $CO_2$ and $CH_4$ production (Figure S4). The calculated $Q_{10}$ values of $CO_2$ production from organic soil were within a narrow range between 4.6 and 5.0. Mineral soils with lower SOC content showed a wider range of $Q_{10}$ values (from 3.6 to 7.3). Permafrost showed significantly lower $Q_{10}$ than both organic and mineral layers (Table S6). Methanogenesis had much larger variations in estimated $Q_{10}$ values. Organic



soils had a $Q_{10}$ value between 18.5 and 48.1, while in mineral soils and permafrost, the average $Q_{10}$ values were 7.1 and 1.6, respectively. Using $Q_{10}$ values to simulate the temperature dependence of processes might work for $CO_2$ production, but could generate significant errors in predicting $CH_4$ production.

### 3.3 Modeled $CO_2$ and $CH_4$ production using observed parameters

The model performed well in simulating $CO_2$ and $CH_4$ dynamics across a range of moisture and SOC gradients and among different soil types (Figure S5, S6). Variations in gas production among different conditions, including microtopographic features, soil layer, and different incubation temperatures were well captured (Figure S7). The comparisons between modeled

and observed $CO_2$ and $CH_4$ production are shown in Figure 5. The model slightly underestimates $CO_2$ production towards the end of the incubations, but still maintain a good agreement between modeled and observed $CO_2$ production ($R^2$=0.89). The underestimation of $CO_2$ production is likely due to substrate limitations caused by the initial distribution of different carbon pools. Model-predicted $CH_4$ production also showed good agreement with observations ($R^2$=0.79). More variation between modeled and observed $CH_4$ production suggests a systematic pattern in the model parameterization of methanogenesis: The

model underestimates $CH_4$ production at 4 and 8°C, and overestimates $CH_4$ production at -2°C.

To assess the model sensitivity to initial model inputs, we compared model predictions in response to varying initial model inputs via perturbation simulations. Significant changes in model predictions were observed in response to perturbations of initial input of SOC, WEOC and moisture (Figure 6). SOC determines the size of different carbon pools in the model, and it further

influences the predictions of WEOC, TOAC, $CO_2$ and $CH_4$, which is consistent with our synthesis results (Table 1). Perturbations in initial WEOC strongly influence the predictions of WEOC, TOAC and $CO_2$, as one of the major assumptions of the model is that the conversion of WEOC to TOAC (fermentation process) is the rate-limiting step. The model also predicted dramatic increases in $CH_4$ and Fe(II) accumulation in response to lower WEOC, which is related to pH response of methanogenesis and Fe(III) reduction. Lower WEOC significantly reduced organic acid accumulation: an increase in pH from

4.92 to 5.15 to 6.14 was predicted in the model when initial WEOC was adjusted 25% or 50% lower, respectively. This drastic pH change is consistent with observations that pH increased up to 1 pH unit during short-term incubations (Roy Chowdhury *et al.*, 2015). Simulated pH increase represented 54% and 354% increases in the value of the pH response function ($f_{pH}$), which substantially increased rates of methanogenesis and iron reduction. Similarly, perturbations in initial soil pH had the strongest effect on the prediction of $CH_4$ by assigning different values in $f_{pH}$ that were directly proportional to the initial reaction rates. The

above results of perturbation simulations demonstrated high sensitivity of this model in response to varying soil geochemical properties.

### 3.4 Model sensitivity to parameterization uncertainties

One of the major assumptions of this modeling framework is to lump multiple fermentation processes into one reaction stoichiometry, controlled by one reaction rate constant. It is critical to evaluate how this simplified structure influences model

performance and contributes to model output uncertainties. The model parameter sensitivity analysis indicated the TOAC pool was most sensitive to changes in the fermentation rate ($R_{fer}$) and reaction stoichiometry (Figure 7). Downstream reactions were less affected by the uncertainties of the two tested parameters. These results supported our assumption of lumped fermentation with fixed stoichiometry, indicating the robustness of the model structure presented here.




A sensitivity analysis on temperature response was performed with four different temperature response functions (Appendix B). In our simulations, the quadratic temperature response function proposed by Ratkowsky et al. predicted much higher $CO_2$ and $CH_4$ production rates at higher temperature, and the lowest rate of both $CO_2$ and $CH_4$ at temperatures below 0 °C, giving the highest temperature response among tested response functions (Figure 8). In contrast, the Arrhenius equation predicted much lower temperature response for both $CO_2$ and $CH_4$. Empirical functions used in CLM-CN and CENTURY gave similar temperature response for both $CO_2$ and $CH_4$. Variations in low temperature $CO_2$ production is well constrained by established temperature response functions, while $CH_4$ production at -2 °C showed a much wider range of temperature response, and the median value is best simulated using Ratkowsky function. This sensitivity analysis is consistent with model output of $CO_2$ and $CH_4$ production, where $CO_2$ is well constrained by the model, but $CH_4$ is significantly overestimated at -2°C using CLM-CN temperature response function. A unified temperature response function for all reactions under different biotic or abiotic constraints substantially contributes to the disagreement between model output and observations.

Redox reactions contribute to proton production or consumption, and the resulting pH alters the value of the pH response function ($f_{pH}$) that directly controls reaction kinetic functions, creating a feedback loop. pH buffering capacity provided by SOM with proton binding sites and pH response function represent two major sources of uncertainties in this feedback loop. Thus, we performed perturbation simulations to characterize the sensitivity of model output to variations in pH buffering capacity and $f_{pH}$ (Figure 9). Higher pH buffering capacity stabilized system pH during prolonged incubations, while lower pH buffering capacity permitted a pH increase by up to 0.71 pH unit compared to the reference simulation. This 14% pH increase led to a 123% increase in $f_{pH}$, accelerating both methanogenesis and Fe(III) reduction rates substantially. Perturbations on pH response function were directly reflected in the slopes of pH response curves (Figure S8). We found up to 372% change in the value of $f_{pH}$ during a 60-day simulation, as steeper increase in $f_{pH}$ accelerated both methanogenesis and iron reduction (equation A2-A5), which contributed to pH rise in the loop, further accelerating $f_{pH}$ increase. Correspondingly, both $CH_4$ and $Fe(II)$ increased by more than 100% after the simulation. While pH buffering capacity is an important factor controlling both redox reactions and pH fluctuations, a unified pH response function for all reactions may impose significant variations in model output.

Soil pH buffering capacity is an intrinsic soil property simulated with simplified linear relationship to soil SOM. However, it generates strong nonlinear response in the simulations of methanogenesis and Fe(III) reduction (Figure 9a). Simulations with varying soil buffering capacity (BC) revealed dynamic pH change at lower BC (Figure 9a, Figure 10c, with BC=1 as reference simulation), and stabilized pH at higher BC. At constant temperature, rates of both methanogenesis and Fe(III) reduction increased significantly at lower BC due to pH control. At lower BC when pH change is not well buffered, higher pH accelerated $CH_4$ and $Fe(II)$ production rates (Figure 10), giving much higher apparent temperature responses, while at higher BC with stabilized pH in the system, apparent temperature responses of these redox processes were significantly lower than the reference simulation (BC=1). Variations in pH buffering capacity generated large variations in apparent temperature responses of methanogenesis and Fe(III) reduction due to the pH feedback loop.

**4 Discussion**

**4.1 Synthesized soil geochemistry and model validation**

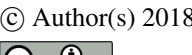


Soil geochemical characteristics represent important abiotic controls on anaerobic carbon decomposition and subsequent $CO_2$ and $CH_4$ production. SOC content, soil pH, water table position, C:N ratio, and landscape position were all suggested to contribute to the variability in anaerobic $CO_2$ and $CH_4$ production (Lee et al., 2012; Schädel et al., 2014; Treat et al., 2015). We synthesized incubation data for gelisol soils from different pedons and soil moisture regimes representing heterogeneity across

the BEO. This coordinated data set allowed us to focus on individual factors and their roles in relation to anaerobic $CO_2$ and $CH_4$ production.

Carbon released as $CO_2$ and $CH_4$ during anoxic incubations decreased with depth. Permafrost was associated with low levels of $CO_2$ production and very low $CH_4$ production, consistent with a previous synthesis (Treat et al., 2015). Nevertheless, permafrost

TOAC, WEOC, and SOC concentrations were all comparable to organic soils, suggesting high substrate availability but low microbial activity. This trend is consistent with previous studies (Walz et al., 2017, Treat et al., 2015), where highest microbial abundance and diversity were observed in surface soil and permafrost contained low microbial abundance (Treat et al., 2014, Waldrop et al., 2010). Among surface soils, higher moisture in LCP soils significantly promoted $CO_2$ and $CH_4$ production and the accumulation of fermentation products (measured as TOAC, Figure 3), emphasizing the importance of soil SOC content and

moisture as strong environmental drivers for carbon decomposition (Table S1, Figure S2). Given the bias in correlation analysis created by the skewed distribution of $CO_2$ and $CH_4$ production in our dataset, additional cluster analysis was performed based on data similarity rather than correlations. High similarity of soil attributes (depth, moisture, pH, C:N ratio, SOC, TOAC) with $CH_4$ production (Figure 11a) was found, suggesting methanogenesis is potentially controlled by a set of soil geochemical characteristics in the local microenvironment.

These synthesized observations support the major assumptions of our model development: (1) the fermentation process converting macromolecular SOM into low molecular weight organic acids is the rate limiting step; (2) different rates of $CO_2$ and $CH_4$ production from different soil layers can be attributed to variations in microbial activity manifested as differences in initial microbial biomass or growth rates. Additional observations of substantial Fe(III) reduction and associated pH increases during

anaerobic decomposition (Figure S9) confirmed the need to simulate pH variations associated with redox reactions and corresponding microbial responses. This anaerobic carbon decomposition framework adequately modulated the involved biotic and abiotic interactions by splitting the carbon flow to different redox reactions and simulating pH buffering capacity to mediate associated changes in acidity or alkalinity.

The model presented here identified fermentation, acetotrophic methanogenesis and acetotrophic iron reduction as key mechanisms for anaerobic $CO_2$ and $CH_4$ production (Vaughn et al., 2016; Lipson et al., 2010). Although denitrification, ammonification and sulfate reduction are all thermodynamically more favorable, low nitrate and sulfate concentrations in BEO soils limit flux through these pathways (Newman et al., 2015). We performed another cluster analysis on the model output (Figure 11b), where we not only simulated fermentation, methanogenesis and iron reduction rates and associated pH changes, but

also tracked the biomass of methanogens (M_Meb) and iron reducers (M_Feb). A dendrogram depicting data similarity showed four distinct clusters comprising of WEOC, $CO_2$ ($CO_2$ prediction), Ferrous (Fe(II) prediction), and $CH_4$ ($CH_4$ prediction) that closely associated with soil geochemical properties and incubation temperature. This result is similar to the cluster analysis of synthesized data, further validating the proposed model structure in capturing major relationships between carbon mineralization and soil geochemical attributes. Predicted $CH_4$ production is strongly influenced by incubation temperature, soil pH, and soil

moisture and depth that determines the size of methanogen population. This model prediction is consistent with previous studies



on the vertical distribution of methanogen population (Waldrop et al., 2010). Environmental factors, such as labile organic matter, water table depth, and soil redox status, soil alkalinity and salinity (Wachinger et al., 2000; Rivkina et al., 2007; Høj et al., 2006; Yang et al., 2017) are all likely to contribute to the variabilities in the distribution and abundance of methanogens and subsequent methane production.

## 4.2 Temperature and pH response of anaerobic carbon decomposition

Rising temperature promotes anaerobic carbon decomposition, resulting in increased rates of anaerobic $CO_2$ and $CH_4$ production (Treat et al., 2014; Lupascu et al., 2012). It is widely recognized that methanogenesis is more sensitive to temperature than respiration (Yvon-Durocher et al., 2014; Yvon-Durocher et al., 2012), and it is usually associated with large variations. Segers estimated the $Q_{10}$ value of methanogenesis ranged from 1.5 to 28 among 1043 incubation experiments using wetland soils
(Segers, 1998). Our data synthesis revealed even higher temperature sensitivity than other reported values. High estimated temperature sensitivity across the freezing point of water has previously been documented (Waldrop et al., 2010) and further attributed to limited water availability for microbial activities at sub-zero temperature (Tilston et al., 2010). Ratkowsky et al. proposed a quadratic relationship for the temperature dependence of microbial growth rates that modeled low-temperature growth better than the Arrhenius Law (Ratkowsky et al., 1982). Our simulations suggest better prediction of methanogenesis
with this temperature response function, possibly due to a more suitable representation of growth limitation of methanogens at sub-zero temperature. Methanogenesis rates are also influenced by the availability of alternative electron acceptors and carbon source. Processes contributing to the accumulation or consumption of carbon substrates and competing electron acceptors may respond differently to temperature change, which could further complicate the temperature sensitivity of methanogenesis. Current modeling approaches heavily depend upon empirical temperature response functions, which may be associated with
large uncertainties due to variations in the selection of data and curve fitting methods. Extrapolation of carbon decomposition rates, particularly methanogenesis rates, into a future warmer climate remains uncertain. More accurate simulations will require additional information on geochemical properties that contribute to the variations of methanogens distribution and methanogenesis activity.

pH values impose fundamental physiological restrictions on microbial activities. Soil pH ranges from acidic to circumneutral (pH 4-7.5) in northern Alaska, and varies substantially through the soil profile and along the microtopographic gradient. Accumulation of organic acids in anoxic soils leads to pH decline (Jones et al., 2003), while consumption of organic acids by methanogenesis and iron reduction increases the alkalinity of the system via the production of $HCO_3^-$ and $OH^-$ (Drake et al., 2015; Roy Chowdhury et al., 2015; Howell et al., 1998). The interplay of these processes leads to strong nonlinear pH feedbacks
in the system, and previous studies have observed up to 1-2 pH unit changes during short-term anoxic incubations (Xu et al., 2015; Drake et al., 2015; Roy Chowdhury et al., 2015). These relationships between pH and organic carbon decomposition can vary in sign and magnitude. Our model simulations with mechanistic pH evolution indicate that constant pH assumed in previous models may cause significant errors in simulating long-term anaerobic $CO_2$ and $CH_4$ production. The intrinsic soil pH buffering capacity plays a large role in stabilizing soil pH and may be heterogeneous depending upon solution acidity or alkalinity, cation
exchange capacity and residual acidity or mineral dissolution. These properties derive from SOM characteristics, moisture, mineral content, and additional geochemical properties, leading to complex correlations between soil pH and SOC decomposition rate that require future investigation.

## 4.3 Fast-decomposing carbon pool



Substrate availability is a primary determinant of potential $CO_2$ and $CH_4$ production (Lee et al., 2012; Schuur et al., 2015; Tarnocai et al., 2009). Total SOC is composed of heterogeneous C pools characterized by different turnover times. Carbon release during short term incubation originates from the C pool with relatively rapid turnover. The size and turnover time of this

quickly-metabolized carbon pool is usually estimated by two-pool or three-pool conceptual models with a maximum likelihood solution using time series of $CO_2$ data (Schädel et al., 2013). A previous study on Siberian permafrost soils using a two-pool model estimated a turnover time of 0.26 years for the fastest-responding pool (Knoblauch et al., 2013). A three-pool model was applied using more extensive incubation datasets collected from 23 high-latitude ecosystems, yielding an estimate of 0.35 years mean turnover time for the fastest-responding carbon pool (Schädel et al., 2014).

In our synthesis study, we directly quantified WEOC and assumed it represented the fast-decomposing labile carbon pool. The size of labile carbon pool is consistent during anaerobic decomposition, while total $CO_2$ and $CH_4$ release represent up to 194% of the labile carbon pool, indicating continuous replenishment of labile carbon pool from non-labile carbon pools within the hierarchy. The replenishment of labile carbon pool can be attributed mostly to decomposition of SOM1 and SOM2 pools with

relatively faster turnover (Phornton & Rosenbloom, 2005). Overall, we estimated the fast-decomposed carbon pool is approximately 2-4% of total SOC, similar to previous estimates. The turnover time calculated from the fermentation rate was comparable to estimates of the turnover time of the fastest-responding carbon pool in previous studies (Figure 12), suggesting these quantifications and parameterization in the anaerobic carbon decomposition framework apply broadly.

**4.4 Key features of the anaerobic model framework and future considerations**

Here we present an anaerobic carbon decomposition framework by combining three well-known modeling approaches developed in different disciplines. A pool-based model to represent upstream carbon transformations and replenishment of DOC pool, a thermodynamically-based model to calculate rate kinetics and biomass growth for methanogenesis and Fe(III) reduction, and a humic ion-binding model for aqueous phase speciation and pH calculation are implemented into the open source geochemical model PHREEQC (Charlton & Parkhurst, 2011). The model framework presented here has several unique features. First, this

model is built upon a thermodynamically-based approach, which allows consistent parameterization of individual reactions along the redox ladder. Such a model structure is particularly useful in circumstances when function-specific microbial growth is difficult to quantify and parameterize. Second, calculations of free energy changes of redox couples are used to modulate redox reaction hierarchy. Considering the difficulty in obtaining growth-associated parameters for every functional group, a thermodynamically-based approach significantly decreases the number of parameters that are difficult to measure. In addition,

proton production and consumption during redox reactions are incorporated into a dynamic pH calculation, allowing various simulations on aqueous solubility and reactivity of different elements. The anaerobic carbon decomposition framework presented here holds a significant advantage over traditional models in simulating carbon decomposition process within a wide range of environmental settings.

In permafrost affected regions, studies consistently identify Fe(III) reduction, denitrification and sulfate reduction (Lipson et al., 2010; Lipson et al., 2013; Ernakovich et al., 2017; Hansen et al., 2007) as alternative anaerobic pathways, which are recognized as energetically more favorable processes than methanogenesis. The new model framework presented here provide a reasonable basis for a deeper understanding of carbon decomposition under oxygen-limited conditions where the importance of accounting for alternative election acceptors becomes more pronounced. Future fine-scale experiments on carbon decomposition using

alternative electron acceptors would be beneficial for more comprehensive parameterization of this model framework. Additional observations on temperature and pH sensitivity of specific redox reactions would also be quite useful in reducing large uncertainties generated by the current representation of temperature and pH responses.

## 5. Conclusion

Microbial processes are the driving forces for biogeochemical cycling of soil carbon, and are subjected to environmental constraints beyond temperature and organic substrate availability. The present study incorporated microbial redox reactions and mechanistic pH evolution to simulate anaerobic carbon decomposition in Arctic soils with depth and across soil moisture gradients. Our data synthesis and modeling results quantify direct effects of temperature on anaerobic carbon decomposition, as well as indirect effects of soil geochemistry that cause strong redox reaction-pH feedback. We identified substantial pH

feedbacks on the predictions on $CO_2$ and $CH_4$ production. The anaerobic carbon decomposition framework presented in this study provided the essential model structure to incorporate redox reactions of alternative electron acceptors for accurate simulation of $CO_2$ and $CH_4$ production. Soil geochemistry impose critical constraints on SOM decomposition, and further regulates permafrost carbon feedback in response to changing climate.

## Code and data availability

PHREEQC is publicly available at http://wwwbrr.cr.usgs.gov/projects/GWC_coupled/phreeqc/.

Input files for model simulation and sensitivity analysis can be found at https://github.com/jianqiuz/decomposition

Data sets used in this work can be found at

http://dx.doi.org/10.5440/1168992
http://dx.doi.org/10.5440/1393836
http://dx.doi.org/10.5440/1288688

## Author contributions

DG, SW, and BG conceived and organized the research study; PT, SP, DG and JZ built the conceptual model framework; JZ
preformed all model simulations; JZ and DG drafted the manuscript. All authors contributed revisions to the manuscript and have given approval to the final version of the manuscript.

## Competing interests

The authors declare no competing interests.

## Acknowledgements

We appreciate comments and suggestions on earlier versions of this paper offered by Ethan Coon. The Next-Generation Ecosystem Experiments in the Arctic (NGEE Arctic) project is supported by the Biological and Environmental Research




program in the U.S. Department of Energy (DOE) Office of Science. Oak Ridge National Laboratory is managed by UT-Battelle, LLC, for the DOE under Contract No. DE-AC05-00OR22725.

**Appendix A: Anaerobic carbon decomposition model**

This section lists reactions used in the anaerobic carbon decomposition model. Under anaerobic conditions, dissolved organic carbon is converted to low molecular weight organic acids via fermentation. One simplified fermentation reaction is used to represent this lumped fermentation process, where 1/3 of the fermented organic carbon is converted to $CO_2$ (Tang et al., 2016;Xu et al., 2015):

$$C_6H_{12}O_6 + 4H_2O \rightarrow 2CH_3COO^- + 2HCO_3^- + 4H^+ + 4H_2 \qquad (A1)$$

This reaction generates protons and decreases pH in the system. Fermentation products acetate and $H_2$ are further consumed via methanogenesis (A2, A3) and iron reduction (A4, A5). The implementation of these reactions in PHREEQC was based on the
bioenergetics of electron donor half reaction, electro acceptor half reaction and cell synthesis to account for biomass growth of methanogens and iron reducers during the reactions.

$$CH_3COO^- + H_2O \rightarrow CH_4 + HCO_3^- \qquad (A2)$$

$$HCO_3^- + 4H_2 + H^+ \rightarrow CH_4 + 3H_2O \qquad (A3)$$

$$CH_3COO^- + 8Fe^{3+} + 4H_2O \rightarrow 8Fe^{2+} + 2HCO_3^- + 9H^+ \qquad (A4)$$

$$2Fe^{3+} + H_2 \rightarrow 2Fe^{2+} + 2H^+ \qquad (A5)$$

In addition, Fe(III) was calculated based on the dissolution of representative amorphous ferric hydroxides (A6), which contributed to pH increase.

$$Fe(OH)_{3(s)} \leftrightarrow Fe^{3+} + 3OH^- \qquad (A6)$$


**Appendix B: Temperature and pH response functions**

We used the CLM_CN temperature response function (B1) in our simulations (Thornton and Rosenbloom, 2005). Additional temperature response functions tested here including B2 used by CENTURY model (Parton et al., 2001), Arrhenius equation B3
used in ecosys (Grant, 1998), and the quadratic equation B4 (Ratkowsky et al., 1983). $T_{ref}$ is set at 25 °C, $E_a$ is the activation energy (J mol$^{-1}$), R is the universal gas constant (J K$^{-1}$ mol$^{-1}$). $T_m$ used in Ratkowsky model represents the conceptual temperature of no metabolic significant, and is set at -8 °C in this study.

$$\ln f(T) = 308.56 \times (\frac{1}{71.02} - \frac{1}{T - 227.13}) \qquad (B1)$$

$$f(T) = 0.56 + 0.465 \arctan [0.097(T - 15.7)] \qquad (B2)$$

$$f(T) = e^{\frac{-E_a}{R}\left(\frac{1}{T} - \frac{1}{T_{ref}}\right)} \qquad (B3)$$

$$f(T) = \left(\frac{T - T_m}{T_{ref} - T_m}\right)^2 \qquad (B4)$$



The bell-shaped pH response function from DLEM model was used here (equation B5, Tian et al., 2010)

$$f(pH) = \frac{1.02}{1.02 + 10^6 \exp(-2.5pH)} \quad (0 < \text{pH} < 7) \tag{B5}$$

$$f(pH) = \frac{1.02}{1.02 + 10^6 \exp(-2.5(14-pH))} \quad (7 < \text{pH} < 14)$$

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



**Table 1. Descriptive statistics and correlation matrix for soil geochemical characteristics, labile carbon pool (in µmol g$^{-1}$**

5   **C) and estimated 60 days max production of CO$_2$ and CH$_4$ (in µmol g$^{-1}$ C) at 8 and -2°C.**

| | 1 | 2 | 3 | 4 | 5 | 6 | 7 | 8a/8b |
|---|---|---|---|---|---|---|---|---|
| 1. SOC | | | | | | | | |
| 2. WEOC | 0.80[a] | | | | | | | |
| 3. TOAC | 0.62[b] | 0.69[a] | | | | | | |
| 4. Moisture | 0.69[a] | 0.82[a] | 0.78[a] | | | | | |
| 5. pH | -0.30 | -0.15 | -0.14 | -0.11 | | | | |
| 6. C/N ratio | 0.07 | 0.06 | 0.17 | 0.05 | -0.64[b] | | | |
| 7. Fe(II) | 0.06 | 0.09 | 0.15 | 0.04 | -0.35 | -0.03 | | |
| 8a. Max_8_CO$_2$ | 0.38 | 0.39 | 0.33 | 0.63[b] | -0.09 | -0.13 | -0.33 | |
| 8b. Max_2_CO$_2$ | 0.40 | 0.54[b] | 0.67[a] | 0.79[a] | 0.07 | -0.14 | -0.30 | |
| 9a. Max_8_CH$_4$ | 0.31 | 0.41 | 0.50 | 0.74[a] | -0.24 | 0.18 | -0.29 | 0.88[a] |
| 9b. Max_2_CH$_4$ | -0.33 | -0.24 | -0.08 | 0.06 | 0.19 | -0.31 | -0.32 | 0.35 |

Note: [a] correlation is significant at the 0.01 level (two-tailed); [b] correlation is significant at the 0.05 level (two-tailed)





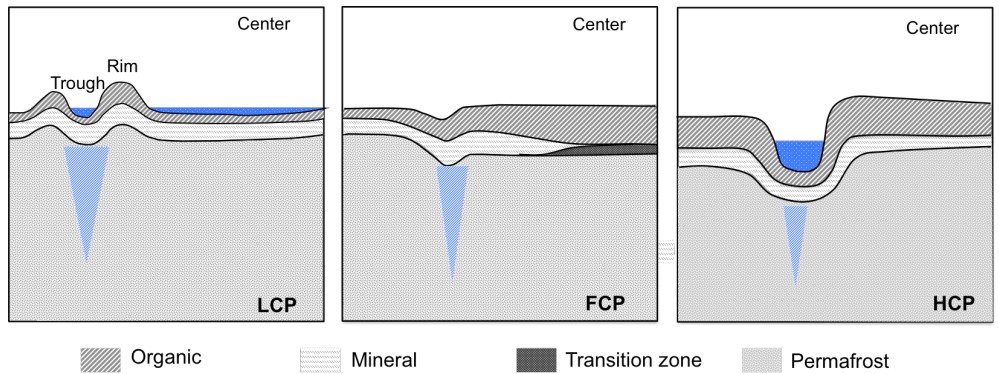

Figure 1. Schematic diagrams of different polygon types and features. The cross section represents the relative landscape
positions of soil profile, including organic, mineral, transition zone and permafrost.





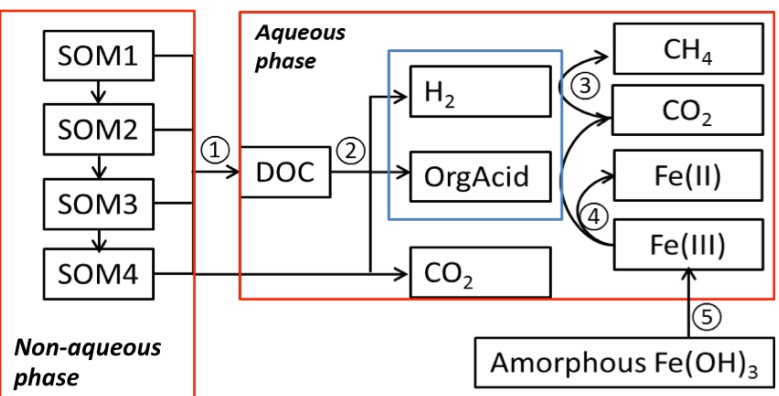

5 **Figure 2. Conceptual diagram showing key processes in the anaerobic carbon decomposition framework. The numbers indicate different processes: 1. SOM degradation from soil organic carbon pools with increasing turnover time produces dissolved organic carbon (DOC) and $CO_2$; 2. Fermentation of DOC into organic acids, $H_2$ and $CO_2$; 3. Methanogenesis from organic acids or $H_2$; 4. Fe(III) reduction from organic acids or $H_2$. 5. Fe(OH)$_3$ dissolution.**





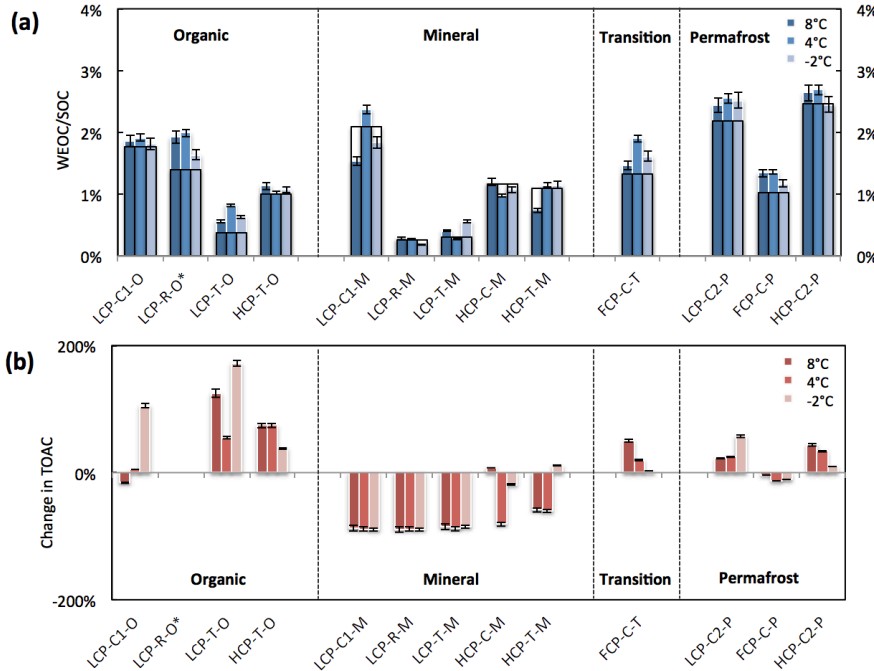

**Figure 3. Changes in (a) WEOC/SOC (quotient of water extractable organic carbon to total soil organic carbon) and (b)**
**TOAC (calculated as (TOAC$_{after}$ - TOAC$_{before}$)/TOAC$_{before}$) after anaerobic incubations at -2, 4 and 8 °C. Bars framed**
**with black lines in panel (a) represent the TOAC/WEOC levels before incubation, and blue bars represent levels after the**
**incubation at corresponding temperatures. Error bars represent standard deviations among triplicate measurements.**





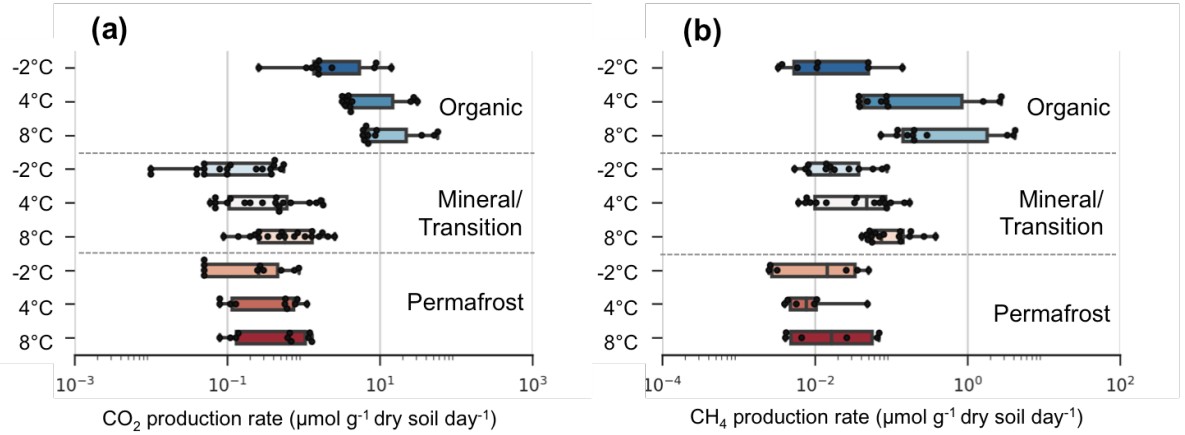

**Figure 4. Box plots show temperature effects on (a) CO$_2$ and (b) CH$_4$ production rates grouped by soil layer. Samples in the transition zone from FCP were pooled with other mineral soils. The two ends of the box represent the 25$^{th}$ and 75$^{th}$ percentile and the lines extending from the box are the 10$^{th}$ and 90$^{th}$ percentile. Please note rates are plotted on log scales.**





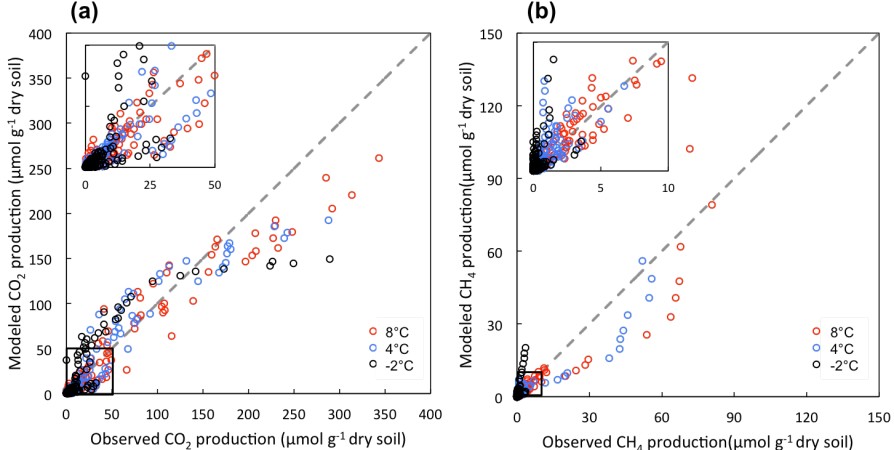

5    **Figure 5. Comparison between modeled and observed production of $CO_2$ (a) and $CH_4$ (b). Averaged measurements of triplicate microcosms at each time point from each incubation temperature were calculated as observed values.**



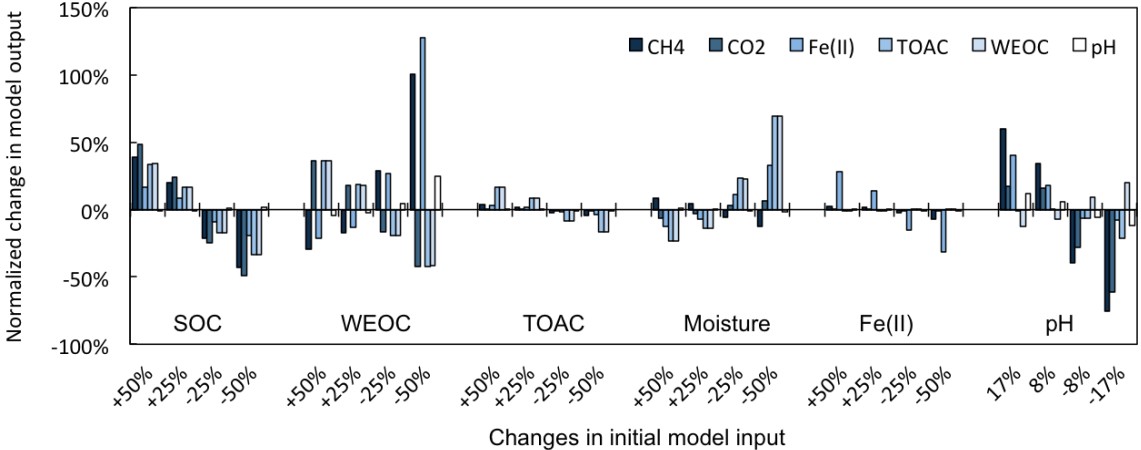

5  **Figure 6. Perturbations of initial soil geochemical conditions differentially affected model predictions (including CH$_4$,**
**CO$_2$, Fe(II), TOAC, WEOC, and pH) during anaerobic carbon decomposition. Normalized changes in model output were**
**calculated as the ratio of perturbation simulation output to reference simulation output after 60 days of anaerobic**
**decomposition at 8 °C. To test model sensitivity in response to initial pH, the reference run started with pH 6, and up to 1**
**pH unit changes was applied in perturbation simulations to represent a realistic pH range for soils. Reference simulations**
10  **were based on soils with 30% SOC (water content=2 g g$^{-1}$ dwt, and pH=5).**




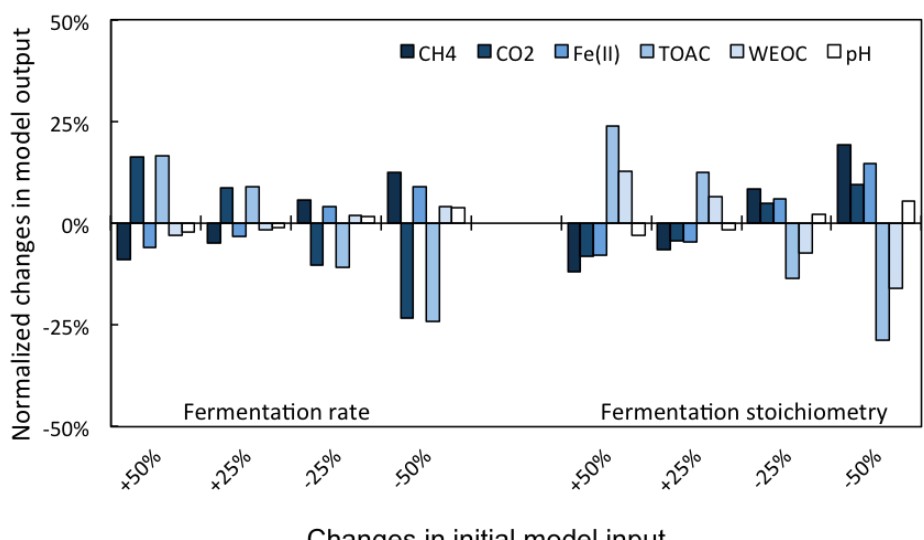

**Figure 7. Simulated changes in model predictions (including CH₄, CO₂, Fe(II), TOAC, WEOC, SOM1, SOM2 and pH) during anaerobic carbon decomposition in response to perturbations of (a) fermentation rate and (b) fermentation stoichiometry (Acetate:CO₂=1:1 for reference simulation). Normalized changes in model output were calculated as the ratio of perturbation simulation output to reference simulation output after 60 days of anaerobic decomposition at 8 °C.**

10  **Reference simulations were based on soils with 30% SOC (water content=2 g g⁻¹ dwt, and pH=5).**





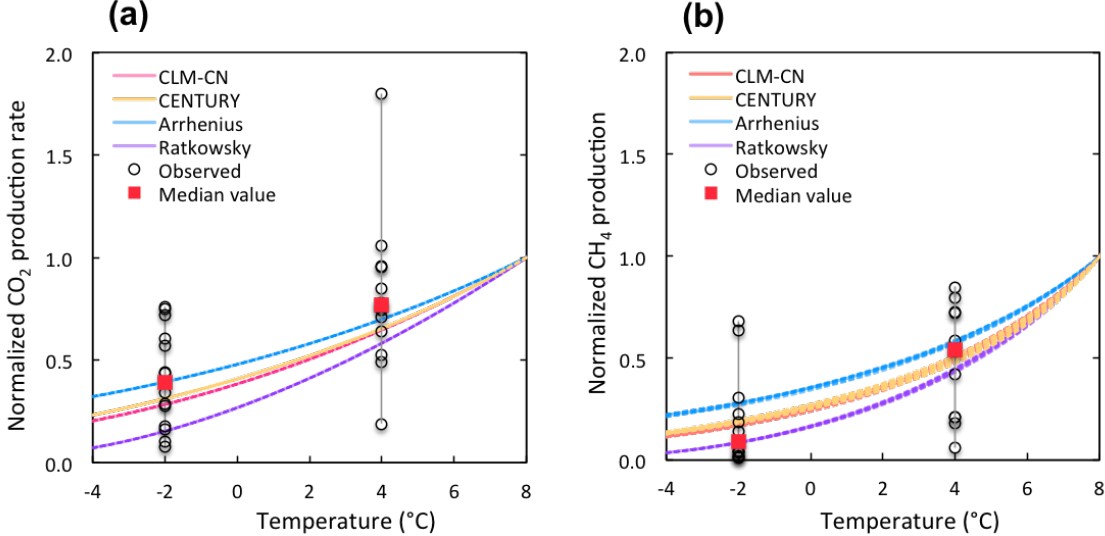

5  **Figure 8. Comparison of simulated and observed temperature response for the production of $CO_2$ (a) and $CH_4$ (b). Results were all normalized to $CO_2$ or $CH_4$ production rates at 8 °C for direct comparison. Observations at -2°C and 4 °C were plotted in black dots and the median value were marked in red. The shaded area represents output uncertainties generated from rate estimations within 60±5 days. Reference simulations were based on soils with 30% SOC (water content=2 g g$^{-1}$ dwt, and pH=5).**



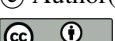

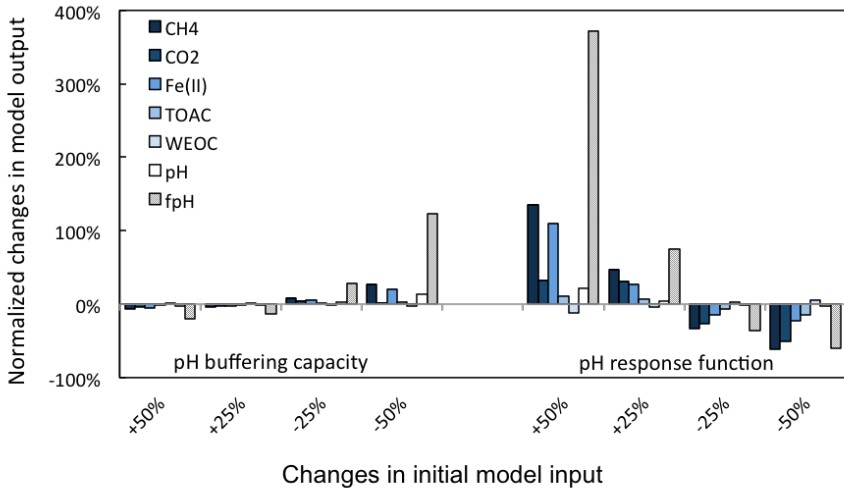

5   **Figure 9. Simulated changes in model predictions (including CH$_4$, CO$_2$, Fe(II), TOAC, WEOC, pH and $f_{pH}$) during anaerobic carbon decomposition in response to perturbations of (a) pH buffering capacity, and (b) pH response function. Normalized changes in model output were calculated as the ratio of perturbation simulation output to reference simulation output after 60 days of anaerobic decomposition at 8 °C. Reference simulations were based on soils with 30% SOC (water content=2 g g$^{-1}$ dwt, and pH=5).**





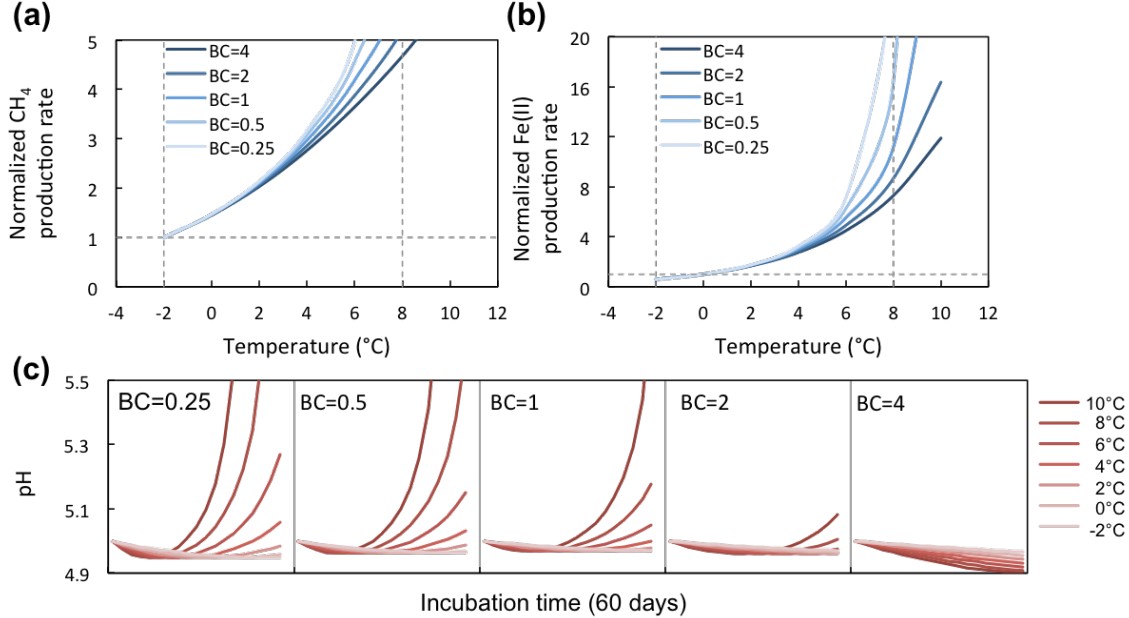

**Figure 10. Temperature response of CH$_4$ and Fe(II) production rates at varying soil pH buffering capacities (BC).**
5    **Varying BCs with respect to reference simulation (BC=1) creates strong feedback to rates of methanogenesis and iron reduction. Reference simulations were based on soils with 30% SOC (water content=2 g g$^{-1}$ dwt, and pH=5).**





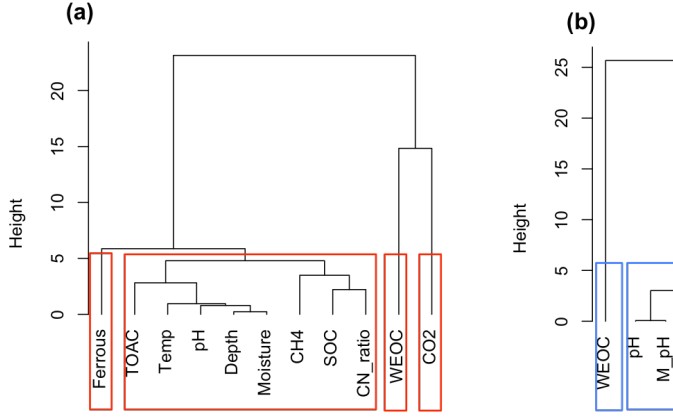

**Figure 11. Cluster analysis of soil geochemical properties related to CO₂ and CH₄ production using Ward's linkage method. (a) cluster analysis of measured soil geochemical characteristics and observed CO₂ and CH₄ production (*n*=42); (b) cluster analysis of modeled results (*n*=42). Model simulated CO₂, CH₄, and Fe(II) production and final pH are labeled as M_CO2, M_CH4, M_Fe, and M_pH, respectively. Biomass of methanogens and iron reducers were tracked in the model and labeled as M_Meb and M_Feb, respectively.**





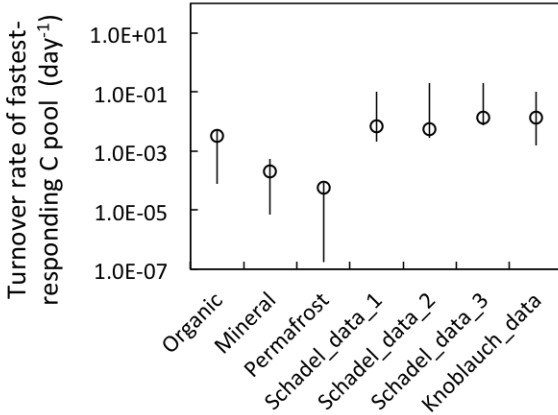

5 **Figure 12. Model estimated turnover rates of the fastest-decomposing carbon pool. Organic, Mineral, and Permafrost labels represent estimations from our model simulations (Rate estimated at 4 °C). Schadel_data represent turnover rates estimated via a three-pool model from pooled anaerobic incubations with normalized incubation temperature of 5 °C (tag 1, 2, and 3 represent pool estimation from different soil types: 1. Organic, 2, Mineral <1m, 3. Mineral >1m) . Knoblauch_data are rate estimates (at 4 °C) made via a two-pool model (Schädel et al., 2014;Knoblauch et al., 2013).**

10 **Open symbols represent the average values, and the vertical lines represent the estimated range.**

