# Peer review of "Modeling anaerobic soil organic carbon decomposition in Arctic polygon tundra: insights into soil geochemical influences on carbon mineralization"

_Biogeosciences, 2018_

## Referee Comment (RC1) · A. Ebrahimi (Referee) · 14 Mar 2018

The manuscript proposes a new model to study organic matter decomposition under anaerobic conditions from arctic soil with a focus on implementing the effects of temperature and pH. The research direction is of a great importance and the authors attempt to formulate such effects on carbon decomposition from arctic is also interesting. However I believe the representation of the manuscript could be significantly improved. My major concern is that the current form of presenting the manuscript is not self-standing and a lot of refers has been done to authors previous publications that makes it difficult to follow and evaluate the content efficiently. Similarly, model description is not complete and no clear hierarchy of the model development and formulation is provided. I understand that the main model has been developed previously but this should not lead to a discontinuous representation that will be non-informative for audience with different background. I strongly suggest improving the model representation and at minimum including a clear schematic with explicit steps that should be taken in formulating such model. Other comments: - Introduction was well-written and provided important and necessary information. However I would still encourage authors to try shortening it that would be focused on the main message of the paper. - While authors acknowledge the key role of hydrolysis to convert SOM (particulate organic matter) to DOC, they have simply ignored this step and no discussion is provided on how the step 1 (Figure 2, conversion of SOM to DOC) is modelled and if hydrolysis is taken into account in the current model. - How the model deals with large discharge rate of DOC that is common in permafrost soil due to lateral flow? - More explanation on how fermentation step is formulated in the model would be helpful. - Q10 values are represented as soil layer combinations. Was there no effect of soil layer? Or there is a correlation with soil depth? More explanation would be helpful. - More explanation on how parameterization has been done and how it has been used in the current model would be nice. - In schematic Figure 2, it is shown that conversion of SOM to DOC produces CO2-? What is the process for this production? Is it general? - Representation of Table 1 should be improved. Is table 1 and Table S4 representing different system? Please be clear in the captions of the Tables. - In the text, it is mentioned that "The maximal production of CO2 is about 2/3 of the initial carbon." Where this number came from? - What "process rich carbon decomposition model" mean? - In Figure 8, could you also show the data at 8C which other data points are normalized with? Is only two data points enough to make a conclusion? How do you illustrate huge variations in observation data? What are the actual values for CO2 and CH4 production rates at 8C? is it for observations? Is absolute data are comparable? Where is the Shaded area mentioned in the caption? - In Figure 3, notations for Figure 3a are not clear. For example LCP-C1-0? - In general, I found it difficult to follow the model results in the form that are represented in Figures

3, 6 and 7, 9. Is there a simpler way of showing the model results that one could extract the trends?

---

## Referee Comment (RC2) · K. Todd-Brown (Referee) · 21 Mar 2018

This paper examines aerobic and anaerobic soil organic matter decomposition in the context of iron, and pH. This is an important contribution to the understanding of soil carbon dynamics in permafrost regions which hold vast reservoirs of carbon that could potentially be released under future climate change. Unfortunately, this manuscript has flow problems with substantial logical gaps between a traditional correlative analysis and the process rich model. More concerning is a lack of documentation on how the process rich model was developed, making the simulation results of this study unreproducible as is.

[Figure]

This paper tries to do both a traditional regression/correlation style analysis and a non-linear process rich simulation. From what I can tell the traditional analysis is solid, although the lack analysis scripts make it difficult to evaluate. However, the connection to the process rich simulation is tenuous at best. In addition, I'm not clear how the data was incorporated into the simulation and how the simulations were validated with the data. I would consider splitting this into two papers, one with the traditional analysis and a second with the model development, parameterization, and validation. While this is not required it would make the manuscripts easier to write. As is there remains work needed on the flow and connection between these two components.

I would like to see some discussion of scaling of these microscale processes to macroscale models.

This study needs a lot more detail to make model development reproducible. The link to github code is a start but documentation is completely inadequate and lack of permanent DOI on the repository means that the codebase might not be there for future studies. The code needs to be commented with major algorithms summarized in functions. README needs instructions on running codebase with a summary of the content of each file. Alternatively this could be submitted as a markdown file with input-function-output format with inline comments explaining approach. Include version number for PHREEQC. Right now, I would not consider this study to be reproducible and it is difficult to evaluate the model results without this context.

I'm concerned that the authors both use a simple correlation analysis to argue for inclusion of various dependent variables in the proposed highly complex non-linear model. In particular, I would not have expected a strong correlation between moisture and SOC given the typical non-linear sensitivity function used to describe respiration response to moisture (though this is possibly explained by the range of moisture conditions considered). In addition, low correlations could be explained by non-linear responses. At the risk of adding yet another analysis to an already confusing study, I would suggest that instead the authors use a paired scatter plot to visually show the relationships between

these variables. This will demonstrate that there is no strong non-linear relationship and that the correlation coefficients are sufficient to describe the relationship.

Line by line reactions:

P1L23 While anaerobic decomposition certainly is missing from many ESMs, I'm not sure I would claim that it is the main driver for model uncertainty. There are several processes which could improve model performance that are currently being investigated and this tripped me up reading through the abstract.

P3L5 Models traditionally do however consider O2 limitation with increasing moisture saturation. I'm almost certain that the authors are aware that traditional moisture sensitivity functions are typically rationalized to have decreasing decomposition under high moisture due to limited O2 diffusion (Orchard and Cook 1983). What this typically does not extend to CH4 emissions, it does implicitly include anaerobic decomposition. A review of implicit vs explicit process representation in decomposition models may be more appropriate here then an outright claim that anaerobic decomposition is not included in ESMs.

P4L28 60days is a short incubation to try to fit a full soils model to. I want to see concerns about time scale addressed somehow here.

P4L38 Why was the 4C dropped form the Q10 calculation??

P5L2 These package citations are less useful without the associated analysis script. Could this please be included in either the SI or as a separate DOI citation?

P5L12 Please be go into more detail on the adaptation of CLM here. Figure 2 is extremely useful but this could use more detail here or in the SI. I would urge the authors to restate the model formulation (even when explicitly drawing on previous work) since frequently it is not clear what portions were modified for the current model. Please include a set of full mathematical equations, descriptions appropriate algorithms, and a fully commented code base used to run the models.

P6L5 Well that is certainly creative model initialization.

P6L22 'further adjusted' Could the authors clarify? Right now it reads as an 'expert tuned' model which is not current best practices given the range of parameter fitting tools that exist.

P6L25 This feels like a very limited sensitivity analysis. An a priori 50% uncertainty seems to be a relatively tight bound for a soil model, especially given the 3 orders of magnitude that was mentioned previously.

P8L7 How was the model calibrated?

P9L18 Was this perturbation analysis done independently of the previous perturbations?

---

## Author Response (AR1)

We appreciate comments from both reviewers and the editor, and have used these to extensively revise our manuscript. This document includes responses to the Associate Editor's comments and reproduces our responses to both reviewers' comments that were previously uploaded. Finally, this document includes a comparison of the revised manuscript with the originally submitted version. These changes were extensive and substantially improved the manuscript's readability. Note that figures were renumbered in the revised version.

**Editorial comments**

*Both reviewers appreciate the modelling efforts conducted and indicate its importance. At the same time though, both indicate to have had major problems with the flow of the story line and with the lack of details on some crucial points of the modelling. Both need to be significantly improved to become acceptable.*
*My suggestion would be to:*
*- eliminate the extensive descriptions of the sites and the performed incubation experiments. These experiments certainly do not constitute the novel part of this manuscript (numerous similar results from similar experiments are available in literature) and it is now distracting (e.g. why extensively discussing Q10 while - I hope- it would be an emergent property from a mechanistic model?). essentials can be moved to suppl. info.*

We moved essential elements of the data synthesis to supplemental text. Our re-analysis of these incubation experiment results required a significant effort, and we hope the product will be useful to future investigators through the dataset that will be available as a DOI.

*- start the methods with a proper model description, possibly using and extended version of fig 2 to guide the flow, then model validation and then model sensitivity. The results section can start with what is now 3.3.*

The revised manuscript includes an extensive description of the model, with the model-data integration illustrated in a new process diagram in Supplementary Figure S2.

*I also have a number of additional comments:*
*- To me, model validation while using a number of initial state var. is quite fine (and is less confusing to me than model initiation), but I don't understand why measured pH was used as parameter in that analysis after the extensive discussion of the authors in the introduction that pH estimates should be based on mechanisms. If you use pH to initialize the model, it does not seem based on mechanisms. Or, to put it more generally: how mechanistic is your model and what is your scientific advance on this topic.? That does nowhere become clear and is also related to my next two comments:*

While geochemical speciation calculations could theoretically estimate pH based on complete measurements of dissolved charged species, this is not practical (or accurate) for Arctic soil water. In our experience, charge balance is usually very poor in these calculations due to the significant contribution of anionic dissolved organic matter, creating large errors in initial pH estimates. This illustrates the importance of the aqueous phase model that we introduce in this work. We recommend usually readily available pH measurements to initialize this model, and our mechanistic pH response function to simulate pH changes from that initial parameter over time.

*- I (and one of the reviewers too) have problems with some claims on the lack of anaerobic decomposition in ESMs. The authors are well aware, I believe, of the ensemble of ESMs developed to model global methane emissions, each of which take anaerobic decomposition processes into account (incl. a model from the CLM family if I am not mistaken). So, how should we see the advance made by this model in this respect?*

The revised manuscript better distinguishes the present explicit representation of anaerobic carbon decomposition cascades and redox processes from previous models that treat processes implicitly. We refer readers to our previous review in this journal by Xu et al. 2016, which provides a more comprehensive description of methane cycle models and a detailed comparison of model structures.

*- The authors next argue that mechanisms should be included and indicate that Eh and pH of models should be*

based on (thermodynamic) mechanisms. Such models are available and extensively used (e.g. in modelling the dynamics in anaerobic sludge reactors, but also for anoxic soil systems) and some of those models are explicitly based on humic acids and/or iron reduction. Moreover, from the current description it does not seem that pH or Eh are prognostic variables in the presented model either. So again: how does the model advance our understanding compared to existing models?

We substantially revised the Introduction and Methods sections to provide more details about this new model. We agree that foundational thermodynamic modeling of anaerobic digestors and batch reactors has benefitted our work, for example by informing the thermodynamic models of Istok, Roden, Bethke et al. cited here.  However, most of these models were developed to simulate specific redox processes including relatively defined carbon substrates. We are not aware of other models that have attempted to couple a thermodynamically based microbial growth model, a substrate pool-based model, and a humic ion-binding model, to create a generic, process-rich carbon decomposition model for anoxic soils that allows simultaneous thermodynamic and pH calculations. Highly parameterized models often perform poorly when used to simulate a wide range of environmental conditions, so the present model is remarkable for its fidelity in simulating CO2 and CH4 production from gradient of soil moisture and SOM conditions.

**Referee #1**

*The manuscript proposes a new model to study organic matter decomposition under anaerobic conditions from arctic soil with a focus on implementing the effects of temperature and pH. The research direction is of a great importance and the authors attempt to formulate such effects on carbon decomposition from arctic is also interesting. However I believe the representation of the manuscript could be significantly improved. My major concern is that the current form of presenting the manuscript is not self-standing and a lot of refers has been done to authors previous publications that makes it difficult to follow and evaluate the content efficiently.*

One of the main goals in this paper (anaerobic model development) is to develop mechanistic representation of methanogenesis, iron reduction and associated pH feedbacks. This goal required data on soil geochemical properties, Fe(III) and Fe(II) concentrations and pH changes during incubations to be synthesized from previous publications. It is not feasible or appropriate to reproduce the high level of detail in those coordinated soil geochemistry measurements, which are described in the cited material. We have moved the synthesis discussion to Supplemental Materials.

*Similarly, model description is not complete and no clear hierarchy of the model development and formulation is provided. I understand that the main model has been developed previously but this should not lead to a discontinuous representation that will be non-informative for audience with different background. I strongly suggest improving the model representation and at minimum including a clear schematic with explicit steps that should be taken in formulating such model.*

This is a good suggestion regarding better model documentation and archiving. To clarify the workflow in the manuscript, we are reorganizing the section that introduces the model and adding a new flow chart in the Supplementary Figure 2 explaining how the synthesized data product is used to inform model development. We are also including a detailed model description in the Supplementary material, supplementing Figure 2 that demonstrates the main structure of the new model. Detailed descriptions of the carbon pool cascade adopted from CLM-CN model, thermodynamically-based growth equations for methanogenesis and iron reduction, and the WHAM model implementation to represent pH buffering will be included in the Supplementary material. Detailed instructions to run the model with our database (redox.dat) under PHREEQC framework will also be included with an example input file. A permanent DOI is reserved for code and additional details on model implementation.

*Other comments:*
*- Introduction was well-written and provided important and necessary information. However I would still encourage authors to try shortening it that would be focused on the main message of the paper.*

We condensed the introduction section to be more focused on introducing explicit processes that are missing from current Earth System Models (ESMs).

*- While authors acknowledge the key role of hydrolysis to convert SOM (particulate organic matter) to DOC, they have simply ignored this step and no discussion is provided on how the step 1 (Figure 2, conversion of SOM to DOC) is modelled and if hydrolysis is taken into account in the current model.*

Hydrolysis is generally recognized as the rate-limiting step during anaerobic carbon decomposition. As stated in the second paragraph in section 2.3, hydrolysis and fermentation include multiple reactions steps, and we combined hydrolysis and fermentation together in one step using an empirical approach. This is a practical assumption for three reasons: (1) Microorganisms that degrade cellulose anaerobically usually also ferment sugars following hydrolysis;
5  (2) Current characterization of SOM in Arctic soils is insufficient to differentiate multiple hydrolysis and fermentation steps: the few reports on Arctic soil exoenzyme activities do not survey the full range of hydrolytic reactions required for biomass decomposition such as endoglycosidases. Thus, data are not available to support multiple hydrolysis and fermentation steps in the model; (3) The lumped hydrolysis and fermentation step still allows us to use this reaction as the rate-limiting step in the model, which fits the observations presented in Figure 3. We will clarify this
10 representation of hydrolysis in the revised manuscript's introduction.

Step 1 (Figure 2, conversion of SOM to DOC) is calculated using the indirect fraction of the original respiration factor from CLM-CN carbon decomposition cascade. The detailed description is included in the commented code file (can be accessed at https://dx.doi.org/10.5440/1430703, once the manuscript revision is finalized). We are also including
15 detailed description in the Supplementary material.

*- How the model deals with large discharge rate of DOC that is common in permafrost soil due to lateral flow?*

20 The Barrow Environmental Observatory is located on the flat Arctic coastal plain, where lateral flow is minimal after snowmelt. Precipitation roughly balances evapotranspiration in most areas during the thaw season. Dealing with lateral flow requires transport processes, which are beyond the scope of current manuscript. However, it is a good target for future research to model different sites, and we are actively working towards coupling PHREEQC capabilities (chemical equilibrium and kinetics) with thermal hydrology models to address transport.
25
*- More explanation on how fermentation step is formulated in the model would be helpful.*
*- More explanation on how parameterization has been done and how it has been used in the current model would be nice.*

30 Additional model details have been added to the Methods section. The fermentation step is parameterized as a single reaction following first order kinetics.

$$C_6H_{12}O_6 + 4H_2O \rightarrow 2CH_3COO^- + 2HCO_3^- + 4H^+ + 4H_2$$

35 As stated in the second paragraph in section 2.3 and first paragraph in section 2.4, the above stoichiometry of fermentation reaction is a lumped process representing production of low molecular weight organic acids (in this case, acetate), $CO_2$ and $H_2$ from labile carbon (we used a constant molecular formula $C_6H_{12}O_6$, representing monosaccharides.

40 Methanogenesis and iron reduction are parameterized using individual growth equations of acetoclastic methanogens, hydrogenotrophic methanogens and iron reducers utilizing acetate or $H_2$. In the revised manuscript, we will include more detailed description on growth equations, and a summary of kinetic rate constants and half saturation constants in the Supplementary material.

45
*- Q10 values are represented as soil layer combinations. Was there no effect of soil layer? Or there is a correlation with soil depth? More explanation would be helpful.*

The initial production rates of $CO_2$ and $CH_4$ used for Q10 calculations showed strong depth effects, as demonstrated
50 in Table S5. Thus we reported the temperature effect (Q10) using grouped soil layers. We further conducted a t-test on the estimated Q10 values for $CO_2$ and $CH_4$ production, respectively (Table S6). These analyses were mentioned in section 3.2. We will add an additional line for clarification in this section.

55 *- In schematic Figure 2, it is shown that conversion of SOM to DOC produces CO2-? What is the process for this production? Is it general?*

In the original CLM-CN carbon decomposition cascade, each carbon pool is associated with a respiration factor representing carbon loss as $CO_2$. Now in our new model, this factor is split into a direct fraction that is respired to
60 $CO_2$, and an indirect fraction that goes to DOC pool. We kept the direct fraction to represent microbial respiration.

$CO_2$ production is also required in anaerobic systems for microbial biomass formation: forming reduced cellular components such as lipids must be offset by $CO_2$ production to balance electrons in the system. In the revised manuscript we will address this briefly in the discussion of Figure 2 and add more detailed explanations in the Supplementary material.

*- Representation of Table 1 should be improved. Is table 1 and Table S4 representing different system? Please be clear in the captions of the Tables.*

10 Gas production data used in the statistical analysis in Table 1 are converted to per gram carbon basis, while in Table S4, data are reported on per gram dry soil basis. The reason we decided to report two tables is due to the high correlations of WEOC, TOAC and soil moisture in respect to SOC. Such correlations conceal the relationships between gas production and other soil geochemical properties. Though it was briefly mentioned in the first paragraph of section 3.2, we will clarify the differences in the revised manuscript and highlight the differences between two
15 tables.

*- In the text, it is mentioned that "The maximal production of CO2 is about 2/3 of the initial carbon." Where this number came from?*

This calculation is based on the stoichiometries of equation A1, A2 and A4. When starting with 1 mol of labile carbon ($C_6H_{12}O_6$), 1/3 of the carbon is released as $CO_2$ during fermentation, and 2/3 of the carbon forms acetate, which can be further respired as $CO_2$ via methanogenesis (1/3 of initial carbon) or iron reduction (2/3 of initial carbon). We will replace this sentence with a more general statement: For complete mineralization, the fraction of initial carbon
25 respired as $CO_2$ is in the range of 2/3-1.

*- What "process rich carbon decomposition model" mean?*

30 That statement means that we explicitly included mechanistic representations of chemical equilibrium processes to allow simultaneous thermodynamic and pH calculations. We will clarify this in the revised manuscript.

*- In Figure 8, could you also show the data at 8C which other data points are normalized with? Is only two data points enough to make a conclusion? How do you illustrate huge variations in observation data? What are the actual values*
35 *for CO2 and CH4 production rates at 8C? is it for observations? Is absolute data are comparable? Where is the Shaded area mentioned in the caption?*

Data at -2 and 4 °C in this figure are normalized to rates measured at 8 °C from corresponding soil samples. i.e. the
40 value for each point at 8 °C is set to 1 in respect to the y-axis scale in the figure. There were 14 averaged observed values for each temperature (each representing a unique soil microtopographic feature × soil layer combination). The absolute values of $CO_2$ and $CH_4$ production rates are plotted in Figure 4 for each temperature. There are huge variations in observations among different soil microtopographic feature × soil layer combinations. We will clarify this interpretation in the revised manuscript.
45

The shaded areas are plotted around each colored line representing model simulations using different temperature response functions. They are quite small, indicating stronger model uncertainties generated from different functions rather then the designated time scale used to run model simulations.

50

*- In Figure 3, notations for Figure 3a are not clear. For example LCP-C1-0?*

The notations of treatment (soil microtopographic feature × soil layer combination) are summarized in Table S1. For example, LCP-C1-O means Low Centered Polygon-Center (the first soil core)- Organic layer. LCP-C2-M means Low
55 Centered Polygon-Center (the second soil core)- Mineral layer. With these notations one can easily identify the microtopographic feature from the figure. We will add an additional line referencing Table S1 for clarity in the revised manuscript.

*- In general, I found it difficult to follow the model results in the form that are represented in Figures 3, 6 and 7, 9. Is*
60 *there a simpler way of showing the model results that one could extract the trends?*

Figure 3 is a bar graph showing changes in WEOC and TOAC pool after incubation. Factors we here include different soil microtopographic feature × soil layer combinations, three different incubation temperatures, and variations among triplicate incubations. All these are essential to demonstrate why we made the model assumption that the DOC pool is in equilibrium state, and the rate-limiting step is the fermentation of DOC into organic acids (Figure 2, process 2).

Figure 6, 7 and 9 are model sensitivity analyses. Variations of ±25% and ±50% were applied to tested parameters (x-axis), the resulting output changes were plotted in bars (y-axis). For example, in Figure 6, when the initial pH is decrease by 8% and 17%, $CH_4$ production decreased by 40% and 80%, respectively. We will provide such an example in the revised text to clarify interpretation.

**Referee#2**

*This paper examines aerobic and anaerobic soil organic matter decomposition in the context of iron, and pH. This is an important contribution to the understanding of soil carbon dynamics in permafrost regions which hold vast reservoirs of carbon that could potentially be released under future climate change. Unfortunately, this manuscript has flow problems with substantial logical gaps between a traditional correlative analysis and the process rich model. More concerning is a lack of documentation on how the process rich model was developed, making the simulation results of this study unreproducible as is. This paper tries to do both a traditional regression/correlation style analysis and a nonlinear process rich simulation. From what I can tell the traditional analysis is solid, although the lack analysis scripts make it difficult to evaluate. However, the connection to the process rich simulation is tenuous at best. In addition, I'm not clear how the data was incorporated into the simulation and how the simulations were validated with the data. I would consider splitting this into two papers, one with the traditional analysis and a second with the model development, parameterization, and validation. While this is not required it would make the manuscripts easier to write. As is there remains work needed on the flow and connection between these two components.*

We appreciate the positive feedback on the synthesis data analysis and the constructive suggestions on making the connections between data synthesis and model development. We moved the data synthesis discussion to the Suplemental materials. To clarify the workflow in the manuscript, we are revising the model introduction (section 2.3) and adding a new flow chart explaining how the synthesized data product is used to inform model development. For example, the bar graph showing changes in WEOC and TOAC pool after incubation (Figure 3) is the motivation to make the model assumption that DOC pool is in equilibrium state, and the rate-limiting step is the fermentation of DOC into organic acids (Figure 2, process 2).

We are also including a detailed model description in the Supplementary material. While Figure 2 demonstrated the main structure of the new model, a revised version of Figure 2 will include the complete CLM-CN carbon decomposition cascade (including the litter pools that currently we are not using in our model) to demonstrate how the carbon pools are adopted from CLM-CN model and modified for our modeling purpose. The new modeling components developed in this work, including thermodynamically-based parameterization for methanogenesis and iron reduction, and WHAM model implementation to represent soil pH buffering, are discussed in great detail in the Supplementary material.

*I would like to see some discussion of scaling of these microscale processes to macroscale models.*

We expanded our current discussion in section 4.4.

*This study needs a lot more detail to make model development reproducible. The link to github code is a start but documentation is completely inadequate and lack of permanent DOI on the repository means that the codebase might not be there for future studies. The code needs to be commented with major algorithms summarized in functions. README needs instructions on running codebase with a summary of the content of each file. Alternatively this could be submitted as a markdown file with input-function-output format with inline comments explaining approach. Include version number for PHREEQC. Right now, I would not consider this study to be reproducible and it is difficult to evaluate the model results without this context.*

We appreciate these suggestions to improve model documentation. Currently the commented model is in the writephrq.py file. To make the model easier to follow, we have added step-by-step instructions on the github site, including PHREEQC installation, how to run the model with our database (redox.dat), and how to create PHREEQC exacutable .phrq files using the python script we have provided. A permanent DOI is reserved for model code and
5  additional details on model implementation is included. We will include example input and output files with detailed comments.  It will be publicly accessible once we finalize the manuscript.

*I'm concerned that the authors both use a simple correlation analysis to argue for inclusion of various dependent*
10 *variables in the proposed highly complex non-linear model. In particular, I would not have expected a strong correlation between moisture and SOC given the typical non-linear sensitivity function used to describe respiration response to moisture (though this is possibly explained by the range of moisture conditions considered). In addition, low correlations could be explained by non-linear responses. At the risk of adding yet another analysis to an already confusing study, I would suggest that instead the authors use a paired scatter plot to visually show the relationships*
15 *between these variables. This will demonstrate that there is no strong non-linear relationship and that the correlation coefficients are sufficient to describe the relationship.*

The correlation between soil moisture and SOC is indeed an interesting result. Measurements of total soil carbon are
20 highly correlated with gravimetric water content in BEO soils (Pearson r = 0.80, P < 0.0001). We will include a paired scatter plot in the revised version. We suggest several alternative explanations. First, high water content in saturated areas preserves organic matter by limiting oxygen diffusion, as the reviewer notes below. Second, undecomposed organic matter binds water tightly, even at low matric potential. Third, high organic matter composition creates large pore volumes that fill with water in saturated soils.

*Line by line reactions:*
*P1L23 While anaerobic decomposition certainly is missing from many ESMs, I'm not sure I would claim that it is the main driver for model uncertainty. There are several processes which could improve model performance that are*
30 *currently being investigated and this tripped me up reading through the abstract.*

We agree that the statement is oversimplified. We will change that to "one of the reasons" driving model uncertainty in saturated soils.

35 *P3L5 Models traditionally do however consider O2 limitation with increasing moisture saturation. I'm almost certain that the authors are aware that traditional moisture sensitivity functions are typically rationalized to have decreasing decomposition under high moisture due to limited O2 diffusion (Orchard and Cook 1983). What this typically does not extend to CH4 emissions, it does implicitly include anaerobic decomposition. A review of implicit vs explicit process representation in decomposition models may be more appropriate here then an outright claim that anaerobic*
40 *decomposition is not included in ESMs.*

Yes, we are aware of the use of moisture functions as a proxy of decomposition level. The suggested term of " implicit vs explicit " is a very nice summary of the problem we were trying to identify in current ESMs. We will summarize implicit vs explicit approaches used in current ESMs to simulate carbon decomposition under anaerobic
45 conditions.

*P4L28 60days is a short incubation to try to fit a full soils model to. I want to see concerns about time scale addressed somehow here.*

50 The length of incubation time was selected because the thaw season in Barrow is about 60-90 days. We have briefly mentioned this in the model development section, as the short incubation is the main reason we adopted the CLM-CN carbon decomposition cascade, since we have no data to fit a full carbon model. We will add additional discussions in both section 4.3 and 4.4 to talk about the limitations of model validation from current datasets and some future considerations.
55

*P4L38 Why was the 4C dropped form the Q10 calculation??*

Originally, we fitted the data from 3 different incubation temperatures. There was no significant difference between the Q10 values estimated by two approaches. We added an additional line to clarify.
60

*P5L2 These package citations are less useful without the associated analysis script. Could this please be included in either the SI or as a separate DOI citation?*

We believe the statistical analysis packages listed here are well documented and applied in this project using standard methods. We will clarify this sentence and provide scripts that are essential for model development through online distribution (see below).

*P5L12 Please be go into more detail on the adaptation of CLM here. Figure 2 is extremely useful but this could use more detail here or in the SI. I would urge the authors to restate the model formulation (even when explicitly drawing on previous work) since frequently it is not clear what portions were modified for the current model. Please include a set of full mathematical equations, descriptions appropriate algorithms, and a fully commented code base used to run the models.*

In the revised manuscript, all these will be included in the Supplementary material. A DOI citation will be available for both model code and synthesized data product (can be accessed at https://dx.doi.org/10.5440/1430703, once the manuscript is finalized). A fully commented code base and step-by-step instruction will be provided with example input and output files. Readers will be able to run our scripts from their own computers.

*P6L5 Well that is certainly creative model initialization.*

Thank you!

*P6L22 'further adjusted' Could the authors clarify? Right now it reads as an 'expert tuned' model which is not current best practices given the range of parameter fitting tools that exist.*

This is a good suggestion. We do not have valid data to verify the biomass of specific functional groups. This lack of data is due to the technical challenges we are facing while doing DNA and qPCR based quantifications. That's the main reason we used thermodynamically-based growth equations to build microbial biomass directly into reaction kinetics. However, we still need a starting point of gross microbial biomass estimations, so the values were selected from previous modeling work done in the Arctic regions. We will add additional explanations in the revised manuscript.

*P6L25 This feels like a very limited sensitivity analysis. An a priori 50% uncertainty seems to be a relatively tight bound for a soil model, especially given the 3 orders of magnitude that was mentioned previously.*

The main purpose of sensitivity analysis is to demonstrate the direction and magnitude of changes. We agree that additional sensitivity analysis on these parameters would be helpful. In the revised manuscript, we added additional sensitivity analysis in Figure 7.

*P8L7 How was the model calibrated?*

The model was calibrated by fitting both $CO_2$ and $CH_4$ production data in two separate steps. We are adding a flow chart explaining how the synthesized data were incorporated into model development and validation.

*P9L18 Was this perturbation analysis done independently of the previous perturbations?*

Yes, the perturbation analysis was done independently of previous perturbations.

[revised manuscript text omitted]

Font: Font color: Dark Gray, Border: : (No border), Pattern: Clear (White)

| Page 16: [10] Deleted | Revised | 6/8/18 5:25:00 PM |
|---|---|---|

value was calculated for each condition to further assess the temperature dependency of $CO_2$ and $CH_4$ production (Figure S4). The calculated $Q_{10}$ values of $CO_2$ production from organic soil were within a

| Page 16: [11] Deleted | Revised | 6/8/18 5:25:00 PM |
|---|---|---|

between 4.6 and 5.0. Mineral soils with lower SOC content showed a wider range of $Q_{10}$ values (from 3.6 to 7.3). Permafrost showed significantly lower $Q_{10}$ than both organic and mineral layers (Table S6). Methanogenesis

| Page 16: [12] Deleted | Revised | 6/8/18 5:25:00 PM |
|---|---|---|

, while in mineral soils and permafrost, the average $Q_{10}$ values were 7.1 and 1.6, respectively.

| Page 16: [13] Formatted | Revised | 6/8/18 5:25:00 PM |
|---|---|---|

Font: Font color: Black, Border: : (No border), Pattern: Clear

| Page 16: [14] Deleted | Revised | 6/8/18 5:25:00 PM |
|---|---|---|

| Page 16: [15] Formatted | Revised | 6/8/18 5:25:00 PM |
|---|---|---|

Space Before: 24 pt, Line spacing: 1.5 lines

| Page 23: [16] Deleted | Revised | 6/8/18 5:25:00 PM |
|---|---|---|

+

| Page 23: [16] Deleted | Revised | 6/8/18 5:25:00 PM |
| --- | --- | --- |

+

| Page 23: [16] Deleted | Revised | 6/8/18 5:25:00 PM |
| --- | --- | --- |

+

| Page 23: [16] Deleted | Revised | 6/8/18 5:25:00 PM |
| --- | --- | --- |

+

| Page 23: [17] Deleted | Revised | 6/8/18 5:25:00 PM |
| --- | --- | --- |

+

| Page 23: [17] Deleted | Revised | 6/8/18 5:25:00 PM |
| --- | --- | --- |

+

| Page 23: [17] Deleted | Revised | 6/8/18 5:25:00 PM |
| --- | --- | --- |

+

| Page 23: [17] Deleted | Revised | 6/8/18 5:25:00 PM |
| --- | --- | --- |

+

| Page 23: [18] Deleted | Revised | 6/8/18 5:25:00 PM |
| --- | --- | --- |

*HC*

| Page 23: [18] Deleted | Revised | 6/8/18 5:25:00 PM |
| --- | --- | --- |

*HC*

| Page 23: [18] Deleted | Revised | 6/8/18 5:25:00 PM |
| --- | --- | --- |

*HC*

| Page 23: [18] Deleted | Revised | 6/8/18 5:25:00 PM |
| --- | --- | --- |

*HC*

| Page 23: [19] Deleted | Revised | 6/8/18 5:25:00 PM |
| --- | --- | --- |

+

| Page 23: [19] Deleted | Revised | 6/8/18 5:25:00 PM |
| --- | --- | --- |

+

| Page 23: [19] Deleted | Revised | 6/8/18 5:25:00 PM |
| --- | --- | --- |

+

| Page 23: [19] Deleted | Revised | 6/8/18 5:25:00 PM |
| --- | --- | --- |

+

| Page 23: [20] Deleted | Revised | 6/8/18 5:25:00 PM |
| --- | --- | --- |

+

| Page 23: [20] Deleted | Revised | 6/8/18 5:25:00 PM |
| --- | --- | --- |

+

| Page 23: [21] Deleted | Revised | 6/8/18 5:25:00 PM |
| --- | --- | --- |

We used the CLM_CN temperature response function (B1) in our simulations (Thornton and Rosenbloom, 2005). Additional temperature response functions tested here including B2 used by CENTURY model (Parton et al., 2001), Arrhenius equation B3 used in ecosys (Grant, 1998), and the quadratic equation B4 (Ratkowsky et al., 1983). $T_{ref}$ is set at 25 °C, $E_a$ is the activation energy (J mol$^{-1}$), R is the universal gas constant (J K$^{-1}$ mol$^{-1}$). $T_m$ used in Ratkowsky model represents the conceptual temperature of no metabolic significant, and is set at -8 °C in this study.

| Page 23: [22] Formatted | Revised | 6/8/18 5:25:00 PM |
| --- | --- | --- |

Left, Tab stops: 3.5", Centered + 6.5", Left

| Page 23: [23] Formatted | Revised | 6/8/18 5:25:00 PM |

Left, Tab stops: 3.31", Centered + 3.75", Centered + 6.5", Left + Not at 3" + 5.69"

| Page 23: [24] Formatted | Revised | 6/8/18 5:25:00 PM |

Left, Tab stops: 3.5", Centered + 6.5", Left

| Page 23: [25] Deleted | Revised | 6/8/18 5:25:00 PM |

=

| Page 23: [25] Deleted | Revised | 6/8/18 5:25:00 PM |

=

| Page 23: [26] Formatted | Revised | 6/8/18 5:25:00 PM |

Left, Tab stops: 3.5", Centered + 6.5", Left + Not at 0.39" + 0.78" + 1.17" + 1.56" + 1.94" + 2.33" + 2.72" + 3.11" + 3.89" + 4.28" + 4.67"

| Page 23: [27] Deleted | Revised | 6/8/18 5:25:00 PM |

| Page 23: [27] Deleted | Revised | 6/8/18 5:25:00 PM |

| Page 23: [28] Formatted | Revised | 6/8/18 5:25:00 PM |

Indent: First line: 0.5", Tab stops: 3.5", Centered

| Page 27: [29] Deleted | Revised | 6/8/18 5:25:00 PM |

**Table 1. Descriptive statistics and correlation matrix for soil geochemical characteristics, labile carbon pool (in $\mu$mol g$^{-1}$ C) and estimated 60 days max production of $CO_2$ and $CH_4$ (in $\mu$mol g$^{-1}$ C) at 8 and -2°C.**

| | 1 | 2 | 3 | 4 | 5 | 6 | 7 | 8a/8b |
|---|---|---|---|---|---|---|---|---|
| 1. SOC | | | | | | | | |
| 2. WEOC | 0.80[a] | | | | | | | |
| 3. TOAC | 0.62[b] | 0.69[a] | | | | | | |
| 4. Moisture | 0.69[a] | 0.82[a] | 0.78[a] | | | | | |
| 5. pH | -0.30 | -0.15 | -0.14 | -0.11 | | | | |
| 6. C/N ratio | 0.07 | 0.06 | 0.17 | 0.05 | -0.64[b] | | | |
| 7. Fe(II) | 0.06 | 0.09 | 0.15 | 0.04 | -0.35 | -0.03 | | |
| 8a. Max_8_CO$_2$ | 0.38 | 0.39 | 0.33 | 0.63[b] | -0.09 | -0.13 | -0.33 | |
| 8b. Max_2_CO$_2$ | 0.40 | 0.54[b] | 0.67[a] | 0.79[a] | 0.07 | -0.14 | -0.30 | |
| 9a. Max_8_CH$_4$ | 0.31 | 0.41 | 0.50 | 0.74[a] | -0.24 | 0.18 | -0.29 | 0.88[a] |
| 9b. Max_2_CH$_4$ | -0.33 | -0.24 | -0.08 | 0.06 | 0.19 | -0.31 | -0.32 | 0.35 |

Note: [a] correlation is significant at the 0.01 level (two-tailed); [b] correlation is significant at the 0.05 level (two-tailed)

[Figure]

**Figure 1. Schematic diagrams of different polygon types and features. The cross section represents the relative landscape positions of soil profile, including organic, mineral, transition zone and permafrost.**

[Figure]

**Figure 3. Changes in (a) WEOC/SOC (quotient of water extractable organic carbon to total soil organic carbon) and (b) TOAC (calculated as $(TOAC_{after} - TOAC_{before})/TOAC_{before}$) after anaerobic incubations at -2, 4 and 8 °C. Bars framed with black lines in panel (a) represent the TOAC/WEOC levels before incubation, and blue bars represent levels after the incubation at corresponding temperatures. Error bars represent standard deviations among triplicate measurements.**

---

## Author Response (AR3)

Referee #1 (30 Oct 2018 report)

*I would like to thank the authors for their rigorous work on addressing the comments on the early version. The manuscript is now significantly improved and easy to read. I have a minor editorial and two major considerations that will hopefully further enhance the quality of the manuscript.*

We appreciate the reviewer's comments, which have improved this manuscript.

*Major concerns:*
*- Authors have nicely responded to my comment on how hydrolysis process is implemented in the model. I understand that there is lack of data to implement hydrolysis step, independently. reason, authors have merged this step with fermentation process by assuming that hydrolysis is the rate limiting step, supported by observations from rice paddy soil. I was wondering how assumption could be for tundra and organic soil, the relevant systems, for this study. This assumption may contribute significant uncertainty for model predictions, since it regulates storage rate of Carbon. For instance, having microbial uptake as the rate limiting step may lead to substantial discharge and accumulation of oligomers depending on hydrological processes. Overall, carbon turnover rate and GHG emission rates over time.*

As the reviewer notes, data are limited comparing rates of anaerobic hydrolysis versus substrate uptake in tundra soils. If microbial uptake were limiting during incubations, we would expect the accumulation of substrates in soil pore water. In our previous paper (Z. Yang et al. / Soil Biology & Biochemistry 95 (2016) 202) we measured a rapid decrease in reducing sugar and ethanol concentrations in pore water, correlated with the production of $CO_2$, $CH_4$ and organic acid fermentation products. When we added glucose to the depleted samples, gas production rates increased quickly. We interpret this result as a limitation in carbohydrate hydrolysis. We elaborated on this point in the revised manuscript on pages 3-4.

*- The absolute values for many parameters are still not provided. I suggest presenting a table that includes all the parameters with values used or fitted from the simulation.*

We added a new supplementary Table S1 "Model parameter values for reactions A1-A5." Also, the full model code is now available online:

Zheng, J., Thornton, P., Painter, S., Gu, B., Wullschleger, S., and Graham, D. E.: Modeling Anaerobic Soil Organic Carbon Decomposition in Arctic Polygon Tundra: Insights into Soil Geochemical Influences on Carbon Mineralization: Modeling Archive, Accessed at https://doi.org/10.5440/1430703, 2018.

This citation has been added to the manuscript.

*- Minor: There are still numerous typo and grammar errors. I suggest a careful reading of the English.*

We carefully proofread the revised manuscript, incorporating numerous small corrections –particularly in the references.

Referee #2 (6 Nov. 2018 report)
*This study examines an integrated model simulating CO2 and CH4 production in permafrost soils and combines a first order linear decay model with a reaction kinetic simulation to look at effects of pH and fermentation on CO2 and CH4 flux. Overall this rewrite is easier to follow then the original version and model development easier to follow.*

We thank the reviewer for productive comments, which we used to clarify and enhance this manuscript.

The authors still have not addressed how the model was tuned and I feel the model documentation could still be improved. This is, essentially, a model development paper since there is no hypothesis that is address with competing models or simulations scenarios. As such I would strongly suggest a table of equations summarizing the various pools which are tracked in the model associated parameters with how their ranges were derived. Relatedly there are vague statements about model fitting (page 5 line 40 specifically) but no actual algorithms given to reproduce parameterization.

We clarified that model optimization was performed using the least squares method by fitting with the observed CO2 production on p. 5, line 34. A description of various carbon pools that were adopted from a previous version of the CLM model with the decomposition cascade structure and parameters is described in Supplementary material (p.5). Furthermore, the complete model code with implementation instructions, input and output files is now available online for perusal (see DOI above).

Ideally I would also like to see a comparison with previous simpler models to motivate the added model complexity. The authors try to get around this with the correlation analysis but there demonstration of model improvement to give a sense of what the gap in the model-data fit was closed by this new formulation.

To characterize the improvements added to this model we introduced new simulations shown in supplemental Figure S10. Starting with a reference model that lacks new modules for iron reduction or dynamic pH calculation, we computed baseline values for $CO_2$, $CH_4$, TOAC and WEOC production, as well as pH and $f_{pH}$ values. The added complexity of iron reduction interacted closely with dynamic pH calculations to produce simulations that better represented observed gas production and pH changes in the modeled incubations. These results shown in Figure S10 are briefly discussed in the main text (page 11, lines 30-38).

In the end the idea of combining a first order linear decay model with a reaction kinetic approach is intriguing and I'm left wanting more out of this paper.

**Comparison of previous version (JuneRevised) with new version (DecRevised). Bibliography changes not shown due to full replacement.**

[revised manuscript text omitted]